

# Ice nucleating properties of the sea ice diatom *Fragilariopsis cylindrus* and its exudates

Lukas Eickhoff[1], Maddalena Bayer-Giraldi[2], Naama Reicher[3], Yinon Rudich[3], Thomas Koop[1]

[1]Faculty of Chemistry, Bielefeld University, Universitätsstraße 25, 33615 Bielefeld, Germany
[2]Alfred-Wegener-Institut Helmholtz-Zentrum für Polar- und Meeresforschung (AWI), Bremerhaven, Germany
[3]Department of Earth and Planetary Sciences, Weizmann Institute of Science, 76100 Rehovot, Israel

*Correspondence to*: Thomas Koop (thomas.koop@uni-bielefeld.de)

**Abstract.** In this study, we investigated the ice nucleation activity of the Antarctic sea ice diatom *Fragilariopsis cylindrus*. Diatoms are the main primary producers of organic carbon in the Southern Ocean and the Antarctic sea ice diatom *F. cylindrus* is one of the predominant species. This psychrophilic diatom is abundant in open waters and within sea ice, and it has developed several mechanisms to cope with the extreme conditions of its environment, for example the production of ice-binding proteins (IBP) and extracellular polymeric substances, known to alter the structure of ice. Here, we investigated the ice nucleation
activity of *F. cylindrus* using a microfluidic device containing individual sub-nanoliter (~90 µm) droplet samples. The experimental method and a newly implemented Poisson statistics-based data evaluation procedure applicable to samples with low ice nucleating particle concentrations were validated by comparative ice nucleation experiments with well-investigated bacterial samples from *Pseudomonas syringae* (Snomax). The experiments reveal an increase of 7.2 °C in the ice nucleation temperatures for seawater containing *F. cylindrus* diatoms when compared to pure seawater. Moreover, also *F. cylindrus*
fragments show ice-nucleation activity, while experiments with *F. cylindrus* ice binding protein (*fc*IBP) show no significant ice nucleation activity. A comparison with experimental results from other diatoms suggests a universal behaviour of polar sea ice diatoms and we provide a diatom mass-based parameterization of their ice-nucleation activity for use in models.

## 1 Introduction

Sea ice is a two-phase medium, composed predominantly of crystalline ice with embedded liquid channels and pockets
(inclusions) where active life can take place. As seawater freezes, dissolved sea salt ions are segregated from the growing ice lattice and accumulate in liquid brine inclusions, which have a lower freezing point due to their high salinity. Its porous structure makes sea ice a habitat for various organisms and enables life within the liquid brine network. Higher irradiance levels in sea ice when compared to the seawater column represent an advantage for photosynthetically active microorganism populating the pore space (Eicken, 1992). During sea ice formation, most microorganisms from the water column remain
entrapped within the ice or are scavenged by floating ice crystals (Ackley and Sullivan, 1994). Species composition changes





with the aging of ice and the stabilization of the brine channel system (Krembs and Engel, 2001), resulting in a dominance of diatom species producing "sticky" extracellular polymeric substances (EPS) with ice-adhering functions.

The diatom *Fragilariopsis cylindrus* (see Fig. 1) is widespread in polar environments and is one of the predominant species
within Antarctic microbial assemblages. The species thrives within sea ice, where it can be found distributed along the sea ice column (Bartsch, 1989; Garrison and Buck, 1989; Günther and Dieckmann, 2001; Poulin et al., 2011). It is, therefore, considered as an indicator of sea ice extend in paleo-environmental studies for reconstructions of past variations (Gersonde and Zielinski, 2000). *F. cylindrus* is also abundant in the water column, for example in the proximity of the sea ice-edge zone (Kang and Fryxell, 1992; Lizotte, 2001) and in ice-covered waters (Garrison and Buck, 1989). *F. cylindrus* has developed a
range of mechanism for coping with the extreme conditions occurring within sea ice (Mock et al., 2017). One prominent example is the production of so-called ice-binding proteins (IBPs) and of other EPS also found in other diatom species (Wilson et al., 2015; Aslam et al., 2018). *F. cylindrus* produces several IBP isoforms (*fc*IBPs), all of which belong to the broadly extended DUF3494 IBP family (Vance et al., 2019). It was shown that *fc*IBP isoform 11 affects the microstructure, i.e., the shape and size, of ice crystals (Bayer-Giraldi et al., 2011; Bayer-Giraldi et al., 2018). Moreover, EPS offer a protective
environment to *F. cylindrus* in order to cope with the conditions of the sea ice habitat (Aslam et al., 2012a; Aslam et al., 2012b; Aslam et al., 2018). It has been suggested that *fc*IBPs accumulate in EPS and, in contact with the icy walls of brine inclusions, alter the pore space resulting in an increased habitability (Bayer-Giraldi et al., 2011).

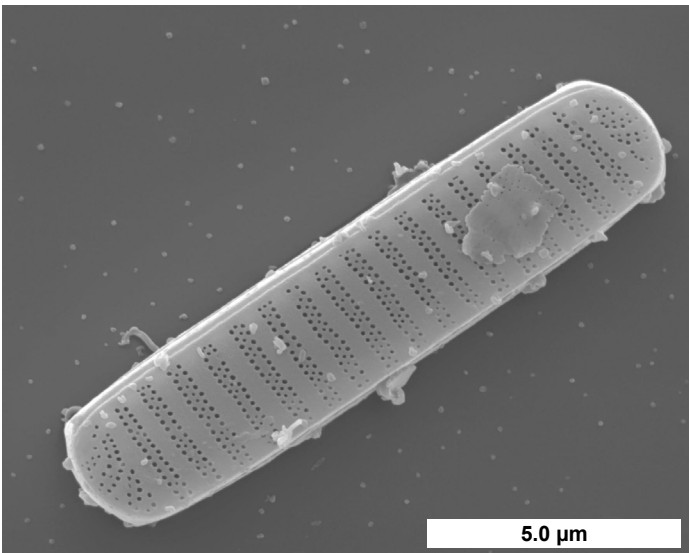

**Figure 1:** *F. cylindrus* cell visualized by scanning electron microscopy. (Image courtesy of Henrik Lange and Friedel Hinz, Alfred Wegener Institute, Germany).

The very good ice-binding properties of *fc*IBP and EPS may imply that the corresponding ice-binding sites could also stabilize the formation of small ice embryos and thereby promote the nucleation of ice from liquid water (Davies, 2014; Bar Dolev et

al., 2016; Eickhoff et al., 2019; Hudait et al., 2019). The rationale behind this proposal is the fact that the active sites for ice

binding and those for the promotion of ice nucleation appear to be quite similar and to match those of crystalline ice (Davies, 2014; Bar Dolev et al., 2016; Eickhoff et al., 2019; Hudait et al., 2019). And indeed, it has been shown both experimentally as well as in molecular dynamics simulations that the ice-binding antifreeze proteins of the mealworm beetle *Tenebrio molitor* (*tm*AFP), which normally prevent the growth of existing ice crystals at temperatures just below 0 °C, can also trigger the nucleation of new ice crystals at lower temperatures (Eickhoff et al., 2019; Hudait et al., 2019). Here, we explore whether a

similar ice-nucleating effect does occur also for IBPs from *F. cylindrus*.

Many particles of biological origin such as bacteria, viruses or diatoms have been detected in the sea surface microlayer as well as in thawing permafrost (Leck and Bigg, 2005; Wilson et al., 2015; Irish et al., 2017; Creamean et al., 2020; Ickes et al., 2020; Roy et al., 2021). Some of these biological particles are able to increase the ice nucleation temperature of small water

droplets and act as ice-nucleating particles INPs (DeMott et al., 2016; Ickes et al., 2020; Welti et al., 2020; Creamean et al., 2021; Hartmann et al., 2021; Roy et al., 2021). These biologic particles can be transported to the atmospheric boundary layer by sea spray aerosol droplets (Irish et al., 2019; Steinke et al., 2022). In the polar atmosphere, they can transported over long distances (Šantl-Temkiv et al., 2019; Šantl-Temkiv et al., 2020). Sea spray aerosol contributes to ice nucleation under mixed-phase cloud conditions as well as at cirrus temperatures in the upper troposphere (DeMott et al., 2016; Hartmann et al., 2021;

Wagner et al., 2021). Further experiments on diatoms and their EPS show that they are able to promote ice nucleation in small droplets of water or seawater (Knopf et al., 2011; Wilson et al., 2015; Ickes et al., 2020; Xi et al., 2021). Thus, diatoms like *F. cylindrus* may have effects on ice nucleation in cloud droplets.

## 2 Material and methods

### 2.1 Sampling and cultivation of the *F. cylindrus* diatoms

The investigated *F. cylindrus* cells belong to the strain TM99 isolated in 1999 from sea ice of the Weddel Sea, Antarctica, by Thomas Mock (*Polarstern* ANT XVI/3 expedition). Stock cultures were kept in f/2 medium (Guillard and Ryther, 1962) set up with Antarctic water and cultivated at 0°C and under continuous illumination of approximately 25 µE m$^{-2}$ s$^{-1}$. Cell numbers were monitored using a Coulter Counter, and cells were harvested during exponential growth phase. Cell cultures were distributed in 50 mL Falcon tubes each containing about 10$^8$ cells, and they were centrifuged at 0°C at 3220 g for 30 minutes.

The clear spent f/2 medium was carefully separated from the cell pellet by pipetting, and both were shock-frozen in liquid nitrogen and stored at -80°C.



## 2.2 Sample preparation

### 2.2.1 Preparation of artificial seawater

For the ice nucleation experiments, we used artificial seawater that mimics the natural conditions in the habitat of Antarctic *F. cylindrus* diatoms. The salinity in the Antarctic region is about 34.5, which corresponds to 34.5 g salts per 1000 g seawater (Roy-Barman and Jeandel, 2016), and we prepared artificial seawater of this salinity for dispersing the diatoms and as a reference for the ice nucleation experiments. For preparing the seawater, the six most important ions were considered, i.e., the cations Sodium, Potassium, Magnesium and Calcium and the anions Chloride and Sulphate, which together make up for about 99.4 % of the dissolved ions in seawater (Roy-Barman and Jeandel, 2016). The required composition was obtained by dissolving 11.8446 g (202.68 mmol) NaCl, 0.3758 g (5.04 mmol) KCl, 5.3280 g (26.21 mmol) $MgCl_2 \cdot 6H_2O$, 4.4902 g (13.94 mmol) $Na_2SO_4 x10H_2O$ and 0.7460 g (5.07 mmol) $CaCl_2 x2H_2O$ in 477.23 g (26.49 mol) double-distilled water. More detailed information on the salts and their concentrations are given in Supplemental Information Table S1. The artificial seawater was filtered through a syringe-filter (0.22 µm, Polyethersulfone, SimplePure) in order to exclude any effect of suspended dust particles on ice nucleation. The samples were stored at a temperature of -18 °C before use.

### 2.2.2 Preparation of *F. cylindrus* samples

The initial *F. cylindrus* samples contained about $10^8$ diatoms per tube, see Sect. 2.1. These samples were placed in a micro reaction tube and were filled up with the filtered artificial seawater to a volume of 2 mL. The resulting stock suspension of $5\times10^7$ cells per mL was used in all experiments. By further dilution with filtered artificial seawater, we generated several more dilute suspensions with concentrations of $1\times10^7$, $2\times10^6$, $1\times10^6$ and $5\times10^5$ cells per mL. For ice nucleation experiments on the fragments and exudates of the *F. cylindrus* cells, we have also filtered these five samples using a syringe-filter (0.22 µm, Polyethersulfone, SimplePure).

In order to identify the ice-nucleating entities of the *F. cylindrus* cells, we separated the different components by means of filtration and centrifugation. We filtered a $2\times10^6$ cells per mL *F. cylindrus* suspension using a syringe-filter (0.22 µm, Polyethersulfone, SimplePure), such that the *F. cylindrus* cells should remain in the filter while smaller fragments of destroyed cells and any soluble species such as soluble ice-binding protein *fc*IBP11 should be able to pass the filter, see Fig. S1 in the Supplemental Information for details. Thereafter, we recovered the filter cake containing the whole *F. cylindrus* cells and larger cell-fragments by shaking the filter in a vial with artificial seawater. Although we used the same volume of artificial seawater as for the preparation of the original cell suspension, we surmise that the concentration of the resuspended diatoms is lower than the initial concentration of $2\times10^6$ cells per mL. Finally, the cell suspension was filtered again (0.22 µm, Polyethersulfone, SimplePure) for comparison with the pure artificial seawater sample. To verify the method all steps were also done with a vial of pure artificial seawater without suspended *F. cylindrus* cells.



We also performed ice nucleation experiments on fresh *f*/2 medium (Guillard and Ryther, 1962) as well as on the spent *f*/2 medium, in which the *F. cylindrus* diatoms were actually grown. The sample preparation procedure is described in detail in Supplemental Information Fig. S2. The spent *f*/2 medium should not contain many cells, because they were separated by centrifugation. Nevertheless, we filtered the medium with a syringe-filter (0.22 μm, Polyethersulfone, SimplePure), such that only small fragments and soluble proteins (e.g., *fc*IBP11) should have remained in the filtrate. In the next step, this sample was

centrifuged using a 100 kDa centrifugal filter (Polyethersulfone, satorius Vivaspin 500, 15000g) such that the remaining solution should not contain any diatom fragments but only smaller soluble molecules such as the soluble *fc*IBP11 protein. For comparison, we also applied the identical centrifugation step with freshly prepared *f*/2 medium that had never been in contact to any diatoms.

### 2.2.5 Preparation of *P. syringae* samples

In additional experiments, we verified our Poisson evaluation procedure (see Sect. 2.3.3). For this purpose, we used well-studied bacterial cells of *P. syringae*, commercially available as Snomax®, from the same batch as investigated in previous studies (Budke and Koop, 2015; Wex et al., 2015). The molecular mass of the individual ice-nucleating proteins in the bacteria is about 150 kDa (Wolber et al., 1986; Govindarajan and Lindow, 1988). A suspension of *P. syringae* with a concentration of 4 mg per mL was prepared from dry Snomax with double-distilled water. By diluting this stock suspension with further double-

distilled water, we also prepared additional more dilute suspensions with concentrations of $1\times10^{-2}$, $2\times10^{-3}$ and $1\times10^{-3}$ mg per mL. Using an average value of the cell number density of $1.4\times10^{9}$ cells per mg (Wex et al., 2015), these mass concentrations correspond to cell concentrations of $1.4\times10^{7}$, $2.8\times10^{6}$ and $1.4\times10^{6}$ cells per mL.

### 2.2.6 Preparation of *fc*IBP11

Previous studies suggest that *fc*IBP11 plays a major role in the response of *F. cylindrus* to freezing conditions (Bayer-Giraldi

et al., 2010), by binding to ice and affecting ice crystal growth (Bayer-Giraldi et al., 2011; Bayer-Giraldi et al., 2018). The *fc*IBP11 protein belongs to the DUF3494 IBP family, which presently constitutes the most broadly spread IBP family often found in sea ice microorganisms (Vance et al., 2019). For our experiments, we used the recombinant *fc*IBP isoform 11 (EMBL Heidelberg), GenBank accession no. DR026070. The protein was expressed as previously described (Bayer-Giraldi et al., 2011) and resuspended in Tris-HCl buffer (pH 7.0). For determining the ice nucleation activity of *fc*IBP11, we prepared a

stock solution with a *fc*IBP11 concentration of 0.1 mmol L$^{-1}$. We diluted this sample by a factor of ten to a concentration of 0.01 mmol L$^{-1}$ using Tris-HCl buffer (pH 7.0) and performed ice nucleation experiments on both sample solutions with the modified WISDOM microfluidic experiment (Reicher et al., 2018; Eickhoff et al., 2019), see below.



### 2.3 Experimental methods for ice nucleation experiments

**2.3.1 Differential scanning calorimetry**

A classic method for the investigation of homogeneous and heterogeneous ice nucleation is differential scanning calorimetry (DSC) of emulsified droplets (Rasmussen and MacKenzie, 1972; Koop, 2004). Here, we used a DSC apparatus (TA-Instruments, DSC-Q100), which was described in detail previously including its calibration procedure (Riechers et al., 2013). As bulk samples notoriously suffer from unwanted impurities, we performed measurements of inverse water-in-oil emulsion

samples containing micrometre-sized droplets. As many thousands of droplets are investigated simultaneously, such samples allow the detection of very reproducible exothermic heterogeneous ice nucleation signals down to the homogeneous ice nucleation temperature of about -38°C (Pinti et al., 2012; Riechers et al., 2013; Dreischmeier et al., 2017).

The principle preparation procedure for the water-in-oil emulsion (w/o) samples was almost identical to the method described earlier (Dreischmeier et al., 2017). 1 mL of 7 wt% emulsifier Span®65 (Merck) dissolved in 93 wt% of a mixture of 50 vol%

methylcyclopenthane (Acros Organics, 99 %) and 50 vol% methylcyclohexane (Acros Organics, 95 %) was used as the organic phase. The aqueous phase consisted of 1 mL of an *F. cylindrus* suspension with a concentration of $1 \times 10^7$ cells per mL, see Sect. 2.2.2 above, or alternatively of 1 mL of pure artificial seawater for comparison. The mixtures of the organic and aqueous phase were subsequently emulsified by stirring with a high-speed disperser (IKA Ultra-Turrax T25 basic) for 10 min at 20'000 rpm. For a DSC measurement, about 10 mg of such an emulsion was filled into an aluminium pan that was sealed hermetically

and then transferred into the calorimeter. The samples were cooled at a rate of -5 °C per min down to -60 °C, and subsequently reheated, first at 5 °C per min and then, in the temperature range between -20 °C and +5 °C, at 1 °C per min.

### 2.3.2 WISDOM microfluidic device

Most of the ice nucleation experiments presented in this study were carried out using droplet microfluidics. In particular, we used a microfluidic device based upon the WISDOM (WeIzmann Supercooled Droplets Observation on a Microarray)

experiment (Reicher et al., 2018), with some minor modifications for a setup operated at Bielefeld University, including adapted temperature and heating rate calibrations, see a previous in-detail description (Eickhoff et al., 2019).

For the droplet generation, we used two syringe pumps (neMESYS NEM-B101-02 E), one filled with the aqueous sample and another with an organic phase consisting of 2 wt% Span®80 (Merck) dissolved in 98 wt% of a mineral oil (Sigma-Aldrich, mineral oil M3516). The microfluidic chip was connected to the pumps with PTFE tubes. The droplets generated within the

chip had diameters of 90 µm ±5 µm.

For the freezing experiments, we placed the microfluidic chip after the droplet production on a temperature-controlled cold-stage (Linkam, BCS 196) attached to an optical microscope (Olympus, BX51 TRF). The temperature of the droplets in the chip was calibrated with respect to the cooling (or heating) rate as well as to the absolute temperature, and is described in detail in a previous study (Eickhoff et al., 2019). The freezing of the droplets was observed using the transmission mode of the

microscope and we recorded the images with a digital camera (Q-Imaging, MicroPublisher 5.0 RTV) for later analysis by a



LabView routine that detects a freezing event from the change in grey values of a particular droplet upon freezing. Typical changes in the droplets' grey values during freezing experiments are depicted in Supplemental Information Fig. S3. In each individual experiment, between about 45 to 70 droplets were observed simultaneously, depending upon the percentage of droplet-filled microcells within the droplet array of the chip.

For all *F. cylindrus* measurements, the chip was first cooled to a temperature of -20 °C at a rate of -5 °C per min, because no freezing events were detected in this temperature range. After equilibration at this temperature for 2 min, the samples were then cooled at a slower rate of -1 °C per min to -45 °C, at which all droplets were frozen. Thereafter, the chip was heated relatively quickly at a rate of 5 °C per min, until -10 °C, and after two minutes of equilibration, it was then heated to 5 °C at 1 °C per min. The detailed temperature profiles for each type of experiment are listed in Supplemental Information Table S2.

**2.3.3 Evaluation procedure for samples with small INP concentrations**

Ice nucleation studies using droplet arrays usually employ high concentrations of INPs, e.g. mineral dust particles or bacterial cells, with a large number of INPs per droplet (Budke and Koop, 2015; Hiranuma et al., 2015; Wex et al., 2015; DeMott et al., 2018; Hiranuma et al., 2019; Kunert et al., 2019; Ickes et al., 2020).

In the present study, the INP concentrations were much lower due to the limited availability of *F. cylindrus* cells. We
investigated droplets with a diameter of 90 μm, corresponding to a volume of about 380 pL. As the concentrations $c$ of *F. cylindrus* cells varied between $5\times10^5$ and $5\times10^7$ cells mL$^{-1}$, the corresponding average INP concentrations were between about only 0.19 up to 19 cells per droplet. It becomes immediately clear that when the average INP concentration $\lambda$ is smaller than 1, i.e. less than one cell per droplet, there must be droplets devoid of any cells, because the number of cells in an individual droplet can only be an integer (assuming only whole cells – without fragments – being present). In such a case, heterogeneous
ice nucleation cannot be triggered in every droplet, but only in those containing at least one cell. Hence, homogeneous ice nucleation is to be expected to occur in the 'empty' droplets. Moreover, even if the average INP concentration $\lambda$ is exactly one per droplet, there will be a few droplets that contain two or more INPs and, thus, other droplets that do not contain any INPs. The distribution of INPs among microfluidic droplets at small average INP concentration can be described using Poisson statistics (Huebner et al., 2007; Köster et al., 2008; Edd et al., 2009; Collins et al., 2015). The following Poisson distribution
can be used to describe the probability $P_\lambda(k)$ that an individual droplet contains exactly $k$ INPs when the average concentration is $\lambda$ INPs per droplet:

$$P_\lambda(k) = \frac{\lambda^k}{k!}\exp(-\lambda) \ . \tag{1}$$

Note that the derivation of the Poisson distribution contains a simplification that require a larger number of droplets and hence Eq. (1) becomes more accurate as the number of investigated droplets increases. For the microfluidic experiments performed
in this work with more than a hundred droplets investigated for each sample the simplification applies.



The average number of INPs per droplet, $\lambda$, is easily calculated from the concertation $c$ of INPs in the stock solution and the volume $V$ of an individual microfluidic droplet:

$$\lambda = c \cdot V_{\mathrm{drop}} . \tag{2}$$

Furthermore, the droplet volume $V_{\mathrm{drop}}$ can be expressed by the droplet's radius $r$ or alternatively by its diameter $d$:

$$V_{\mathrm{drop}} = \frac{4}{3}\pi \cdot r^3 = \frac{1}{6}\pi \cdot d^3 . \tag{3}$$

Figure 2 shows the calculated Poisson distributions of the number of cells per droplet for four different values of $\lambda$ in a concentration range relevant to this study. For lower values of $\lambda$, the histograms exhibit the tilted shape typical of Poisson distributions, while for larger values of $\lambda$, the Poisson distribution approaches the more symmetrical shape of a normal distribution (Koop et al., 1997).

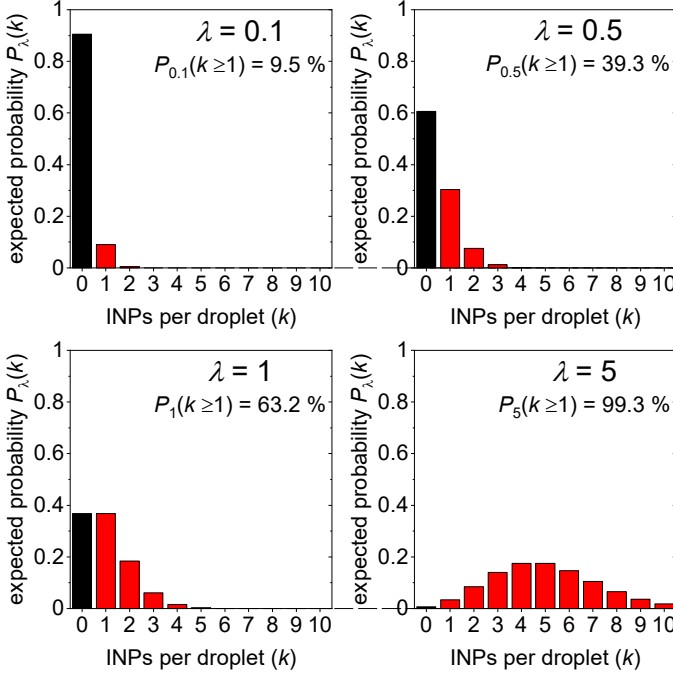

**Figure 2:** Calculated probability $P_\lambda(k)$ of the number $k$ of INPs per droplet for different values $\lambda$ of the average cell concentration per droplet. The black-coloured bars indicate the probability for the occurrence of droplets without any INPs, while the red-coloured bars indicate the combined probability $P_\lambda(k \geq 1)$ for droplets containing at least one INP. The corresponding values for $P_\lambda(k \geq 1)$ are annotated in each panel for different values of $\lambda$.

For the ice nucleation experiments considered here, only those droplets containing at least one INP and those without any INPs are relevant, as this determines whether they are subject to heterogeneous or homogeneous nucleation, respectively. Whether a droplet contains one, two or more INPs is of less importance, as long as every INP is identical and, thus, induces





heterogeneous ice nucleation at the same temperature. The probability that a droplet does not contain any INPs can be

calculated easily by inserting $k = 0$ into Eq. (1):

$$P_\lambda(0) = \frac{\lambda^0}{0!} \exp(-\lambda) = \frac{1}{1} \exp(-\lambda) = \exp(-\lambda) \ . \qquad (4)$$

$P_\lambda(0)$ is shown as the black-coloured bar in each panel of Fig. 2. The probability that a droplet contains at least one

INP, $P_\lambda(k \geq 1)$, is given by the combined probability of all red-coloured bars in each panel of Fig. 2, and it can be calculated

using the fact that the sum of all probabilities $P_\lambda(k)$ for $k$ from 0 to $\infty$ must become 1 (see Eq. (5)):

$$P_\lambda(k) = \sum_{k=0}^{\infty} \frac{\lambda^k}{k!} \exp(-\lambda) = 1 \ . \qquad (5)$$

Hence, $P_\lambda(k \geq 1)$ can be calculated from the following difference

$$P_\lambda(k \geq 1) = \sum_{k=1}^{\infty} \frac{\lambda^k}{k!} \exp(-\lambda) = \sum_{k=0}^{\infty} \frac{\lambda^k}{k!} \exp(-\lambda) - \sum_{k=0}^{0} \frac{\lambda^k}{k!} \exp(-\lambda) = 1 - P_\lambda(0) = 1 - \exp(-\lambda) \ . \qquad (6)$$

Since $\lambda$ can be expressed by the product of the droplets' diameter and the known concentration of INPs in the stock solution,

$c$, (see Eq. (2) and Eq. (3)) this yields:

$$P_\lambda(k \geq 1) = 1 - \exp\left(-\frac{\pi}{6} \cdot c \cdot d^3\right) \ . \qquad (7)$$

The equations above have been derived for applications where the average concentration $c$ of INPs in solution is known.

However, in ice nucleation experiments of natural samples, the concentration $c$ of INPs per volume is often unknown a priori

and other values such as the organic carbon content has to be used for comparison (Gute and Abbatt, 2020; Xi et al., 2021). In

such cases, Eq. (7) can be used to obtain a rough estimate of $\lambda$ and, thus, $c$ from ice nucleation experiments when a plateau in

the experimental frozen fraction curve is observed. The frozen fraction is defined as the number of frozen droplets relative to

the number of all droplets, at a given temperature (Budke and Koop, 2015). Here, we term the value of the frozen fraction at

the plateau as $f'_{ice}$. If a sufficiently large number of droplets is investigated, then $f'_{ice}$ corresponds to the fraction of droplets

that froze heterogeneously and thus may be equated with that fraction of droplets containing at least one INP.

There are two underlying assumptions for experimentally obtaining $f'_{ice}$. First, every droplet containing at least one INP freezes

heterogeneously, which appears entirely reasonable. Secondly, every droplet containing one or more INPs freezes at a higher

temperature than those droplets without any INP, i.e. the difference between the heterogeneous and homogeneous ice

nucleation temperature is large enough to be easily distinguished in the experiment. With these two assumptions a plateau in

the frozen fraction curve can be interpreted as follows: the fraction of droplets below the plateau froze heterogeneously and

contain at least one INP, and the fraction of droplets above the plateau froze homogenously (when their freezing temperature

is consistent with homogenous freezing) and, thus, do not contain INPs. In practise, this evaluation procedure does not work

if none of the droplets froze heterogeneously or if all droplets froze heterogeneously at the same temperature without any





obvious plateau, i.e. it is only applicable for intermediate average INP concentrations in what we term the "Poisson relevant range".

We define this "Poisson relevant range" as the range of average INP concentrations, in which both droplets without any INP as well as droplets containing one or more INPs occur and, thus, both can be observed readily in the corresponding freezing experiments. For the experiments presented here, we establish the Poisson relevant range as the area between $P_\lambda(k \geq 1)$ values of 5.0 % and 99.5 %. The lower limit was set at 5.0% in order to avoid any influence of the freezing of a minor fraction of droplets induced by impurities, the upper limit corresponds to about one out of 200 droplets not containing any INP and thus

freezing homogeneously. For higher concentrations, when every droplet contains at least one INP, the above Poisson evaluation is not needed and the classic method can be used, and so this upper limit sets an endpoint for the Poisson-based evaluation. The classic method indeed assumes that every observed droplet contains at least one INP and it has been described in detail previously (Murray et al., 2012; Budke and Koop, 2015).

To demonstrate the concentration range suitable for the Poisson method, i.e. the Poisson relevant range, the latter is indicated

in Fig. 3a as the grey shaded area. The solid blue curve shows the values of $P_\lambda(k \geq 1)$ calculated using Eq. (7) as a function of the average INP concentration $c$ of the studied sample and a droplet diameter of 90 µm. The two dashed lines show the changes for a deviation of ±5 µm in droplet diameter.

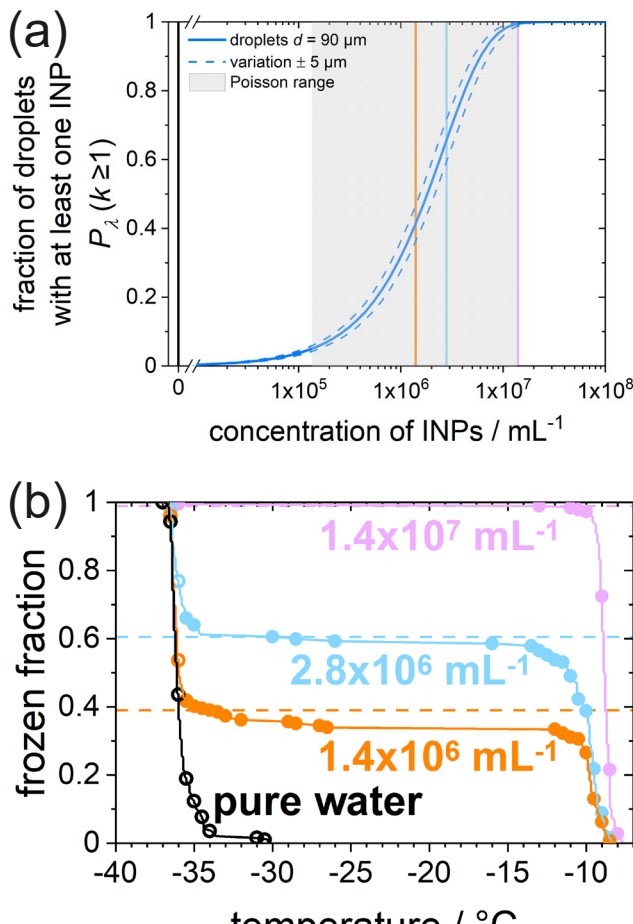

**Figure 3: (a)** Fraction of droplets containing at least one INP, $P_\lambda(k \geq 1)$ as a function of INP concentration in the investigated *P. syringae* sample. The solid blue curve represents the values of $P_\lambda(k \geq 1)$ for the droplets in the WISDOM experiment with a diameter of 90 μm, calculated using Eq. (7), the dashed curves indicate the uncertainty for a variation of ±5 μm in droplet diameter. Eq. The grey shaded area shows the Poisson relevant range, with the lower and upper limits at the concentrations corresponding to $P_\lambda(k \geq 1) = 0.050$ and $P_\lambda(k \geq 1) = 0.995$, respectively. The coloured vertical bars mark the experimentally investigated concentrations of *P. syringae*: $1.4 \times 10^6$ mL$^{-1}$ (orange), $2.8 \times 10^6$ mL$^{-1}$ (blue), and $1.4 \times 10^7$ mL$^{-1}$ (purple) and pure water (black). A comparable plot for the *F. cylindrus* diatoms can be found in Fig. S4. **(b)** Fraction of frozen droplets as a function of temperature for different concentrations of *P. syringae* bacteria in double-distilled water (coloured) and pure double-distilled water (black) for reference. The horizontal lines mark the values for $P_\lambda(k \geq 1)_{measured}$, see text. Data points of frozen fractions are binned in temperature intervals of 0.5 °C (intervals without freezing events are not shown). Filled circles represent droplets containing *P. syringae* cells (based on calculations for $P_\lambda(k \geq 1)$ with Eq. (7)), in which freezing was induced heterogeneously. Open circles represent droplets that should not contain *P. syringae* according to the calculations and, thus froze homogenously.

To verify the procedure, we investigated aqueous suspensions of the well-studied ice-nucleating bacterium *Pseudomonas syringae* in the form of the commercial product Snomax (Morris et al., 2011; Budke and Koop, 2015; Wex et al., 2015). The ice nucleation temperatures of each about 165±15 droplets, from three single measurements with 45 to 70 droplets each,





containing either pure double-distilled water or three different concentrations of *P. syringae* were investigated, see Supplemental Information Table S3. These concentrations are also marked in Fig. 3a as vertical lines. A similar plot for the *F. cylindrus* diatoms can be found in Fig. S4. The resulting experimental frozen fraction curves of *P. syringae* are shown in Fig. 3b. Double-distilled water (black open symbols) shows a steep increase in frozen fraction below about -34.0 °C, in agreement with homogeneous ice nucleation rates of droplets of such diameter (Koop and Murray, 2016; Reicher et al., 2018; Eickhoff

et al., 2019). Following this observation, all droplets of the *P. syringae* samples that froze at around or below this temperature are assumed to have nucleated homogenously, i.e. they are considered to contain no INPs in the analysis below.

   For all *P. syringae* samples, the first freezing events occur at much higher temperatures of about -8 to -9 °C, and the frozen fraction curve in each case initially increases strongly before reaching a plateau, and subsequently the remaining liquid droplets freeze only at very low temperatures. In each sample, the plateau occurs at a different value of the frozen fraction, e.g. $f'_{\text{ice}}$ is

higher the larger the *P. syringae* concentrations (pink > blue > orange). We determined the corresponding $f'_{\text{ice}}$ values, as defined above, from the experimentally obtained frozen fraction curve as the value of the frozen fraction at -34.0 °C, i.e. at the threshold between heterogeneous and homogenous ice nucleation as defined above. The resulting $f'_{\text{ice}}$ values for the three concentrations were 0.99, 0.61, and 0.39, respectively, indicated as the dashed horizontal lines in Fig. 3b. These $f'_{\text{ice}}$ values correspond to $P_\lambda(k \geq 1)_{\text{measured}}$ and can be used to infer the average INP concentration from Eq. (7). Because in the current

experiments the INP concentrations are known (i.e., $1.4 \times 10^7$, $2.8 \times 10^6$. and $1.4 \times 10^6$ mL$^{-1}$), these experimentally derived $f'_{\text{ice}}$ values can be compared to the expected $f_{\text{ice}}$ values, corresponding to $P_\lambda(k \geq 1)_{\text{calcualted}}$ values calculated from Eq. (7), yielding values of $1.00 \pm 0.01$, $0.66 \pm 0.06$, and $0.41 \pm 0.05$, respectively. These theoretical values are in good agreement (within experimental uncertainty) with the measured values and thus confirm our approach and the inferred INP concentrations of $1.2 \times 10^7$, $2.5 \times 10^6$. and $1.3 \times 10^6$ mL$^{-1}$ (see Supplementary Table S3) deviate by about 14%, 11% and 7% from the prepared

concentrations, which is very good given that INP concentrations can vary by orders of magnitude. For further validation that the Poisson distribution is necessary for a proper evaluation in the above-mentioned concentration range, the cumulative number of active ice-nucleating sites $n_N$ per number of *P. syringae* bacteria was evaluated and discussed in the text describing Supplementary Fig. S5.

## 3 Results and Discussion

**3.1 Ice nucleation of *F. cylindrus***

### 3.1.1 Differential Scanning Calorimetry

As an initial experiment, the ice nucleation activity of *F. cylindrus* diatom cells was studied by differential scanning calorimetry. For these measurements, an inverse emulsion of pure artificial seawater as a reference was compared with an emulsion of artificial seawater containing $1 \times 10^7$ *F. cylindrus* cells per mL, see Fig. 4.


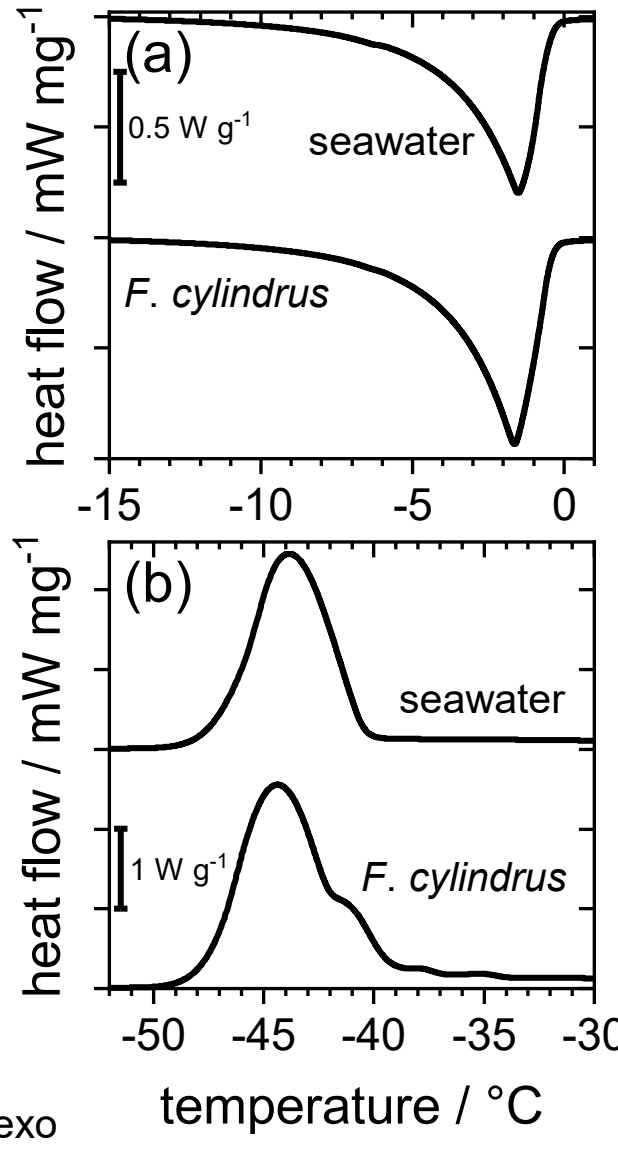

**Figure 4:** Comparison of DSC thermograms of water-in-oil emulsions containing pure artificial seawater and artificial seawater with *F. cylindrus* cells. **(a)** The endothermic melting-signals are almost identical for pure seawater and seawater containing diatoms. **(b)** Exothermic freezing signals for pure seawater and seawater containing diatoms. While the seawater emulsion shows only one freezing signal, the emulsion containing *F. cylindrus* shows the same signal but with a shoulder and smaller signals at higher temperature, indicative of diatom-induced heterogeneous ice nucleation.

First, the endothermic ice melting signals of the reference and the sample in Fig. 4a show almost the same signal, indicating that any colligative effect of the diatoms is negligible when compared to the amount of the dissolved ions in the artificial seawater. This similarity in the ice melting signals also implies no change in water activity of the artificial seawater upon the addition of the diatoms and, thus, no colligative effect on the homogeneous ice nucleation (freezing) signals is to be expected



The exothermic freezing signals for both emulsions are shown in Fig. 4b. For the seawater reference, one distinct nearly symmetrical freezing signal is revealed with a maximum at about -44 °C and an onset at about -40 °C. In contrast, the *F. cylindrus* sample shows the same maximum, but in addition a second exothermic signal in the form of a shoulder at about -42 °C, with an onset at a somewhat higher temperature of -39 °C when compared to the reference, and with small signals as high as -34 °C.

The larger broad signal in both emulsion samples corresponds to the homogeneous ice nucleation temperature of artificial seawater. This signal is also observed in the *F. cylindrus* sample because many of the emulsion droplets in that sample do not contain diatoms. The exothermic shoulder of the signal, which is not present in the reference, is most likely due to the freezing of droplets containing a diatom cell or fragment, and the shift of the onset to higher temperature is a first indication for the heterogeneous ice nucleation activity of the diatoms.

Because of the fact that the diatoms are of similar size as the emulsion droplets and the potential of mechanical disruption of diatom cells during the fast stirring of the disperser during emulsion preparation, these emulsion experiments appear to us as not suitable for a quantitative analysis of the ice nucleation activity of *F. cylindrus*. Thus, we employed non-invasive methods in the experiments described below.

### 3.1.2 Droplet Microfluidics

First, we investigated the ice-nucleating properties of samples containing *F. cylindrus* diatoms at different concentration suspended in artificial seawater. For this purpose, we made use of the droplet microfluidic devices described in Sect. 2.3.2 above. The results of these experiments are presented in Fig. 5a, which shows, as a function of temperature, the frozen fraction of droplets $f_{ice}$, commonly defined as the cumulative number of droplets frozen when cooled to a certain temperature relative to the total number of droplets (Murray et al., 2012). Thus, $f_{ice}$, is practically independent of the total number of droplets

investigated in a particular experiment. In our case, the number of droplets varied between 45 and 70 droplets per single measurement, and typically three single measurements per sample were performed. Figure 5a shows that the freezing temperatures of all *F. cylindrus* samples (coloured symbols) are higher than that of the artificial seawater reference sample (grey symbols), hence supporting the observations from the DSC experiments above that the *F. cylindrus* diatoms promote ice nucleation. To compare the different samples, we use the $T_{50}$ temperature, which is defined as that temperature at which half

of the observed droplets are frozen, i.e. $f_{ice} = 0.5$. For the artificial seawater, we measured a $T_{50}$ of -40.1 °C, and $T_{50}$ of the *F. cylindrus* suspensions is shifted to higher temperature by about 2.8 °C to 7.2 °C with increasing diatom concentration. Detailed information on the increase in $T_{50}$ of the different concentrations is given in the Supplemental Information Table S4. This significant concentration dependence of the $T_{50}$ shift reveals that not all diatoms nucleate ice at exactly the same temperature and implies a distribution of the ice nucleation efficiency as has been observed previously also for other ice

nucleators (Herbert et al., 2014; Budke and Koop, 2015).

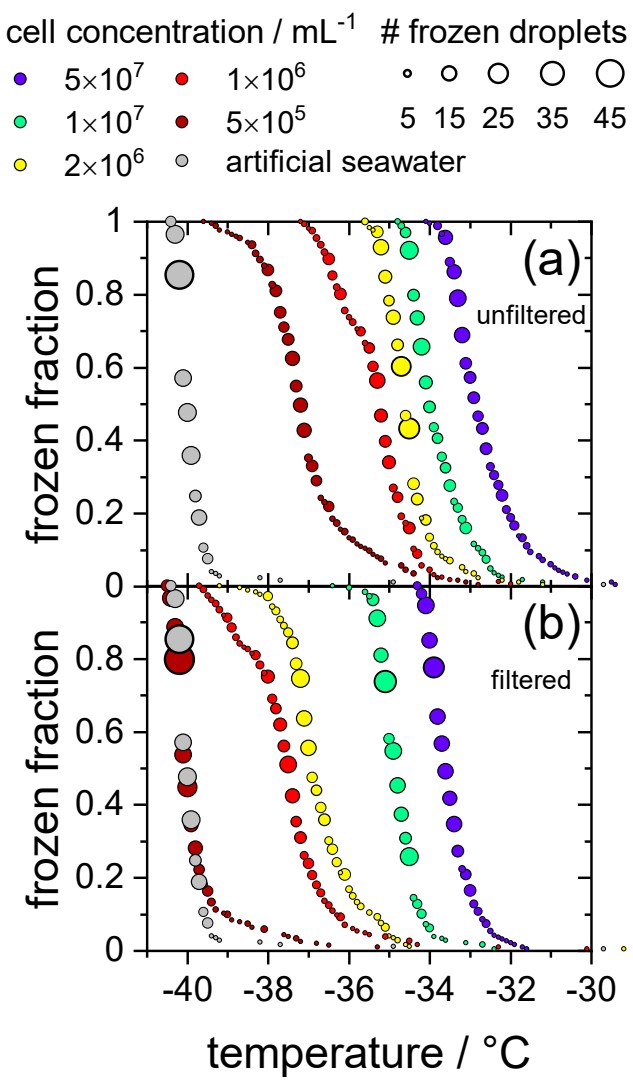

**Figure 5:** Cumulative fraction of frozen droplets as a function of temperature for different *F. cylindrus* concentrations (coloured circles) and pure artificial seawater (grey circles) as a reference. The size of the circles indicates the number of droplets frozen within the same temperature interval (0.1 °C). Each dataset combines three individual measurements containing each between 45 and 70 droplets. **(a):** Frozen fraction curves for the five *F. cylindrus* samples, containing mostly whole diatoms and, probably, some fragments. **(b):** Freezing temperatures of the same samples shown in panel **(a)**, but after filtration with a pore size of 0.22 μm. These samples, thus, contain no whole cells but fragments as well as proteins and other soluble components. Note that the concentrations refer to the diatom concentrations before filtration. The seawater reference (grey circles) is the same in both panels.

This fact can be visualized better by plotting the cumulative number $n_N$ of ice nucleating sites per number of *F. cylindrus* diatoms, defined in Eq. (S2), as a function of freezing temperature, see Fig. 6. This $n_N$ value is independent of the concentration of investigated INP and of the size of the investigated droplets, but can be measured for a wide range of temperatures using different concentrations, and allows for the comparison with results from other experimental techniques (see discussion below). Figure 6 reveals that at -30.0 °C ~0.1 % of the *F. cylindrus* diatoms promote ice nucleation, which increases to ~1 % at -32.0



°C and ~10 % at -33.5 °C. Between about -35.0 °C and -36.5 °C all diatoms trigger the nucleation of ice, i.e. $n_N = 1$. By

definition, $n_N$ values larger than one should not be possible, because it would imply that one diatom can induce the freezing

of more than one droplet, which is unreasonable. The highest $n_N$ values occur at the lowest diatom concentrations and,

therefore, we must consider the Poisson range defined above, i.e. whether or not each droplet does indeed contain a diatom

cell. Following the treatise in Sect. 2.3.3 using Eq. (7), we indicate in Fig. 6 all droplets that contain at least one diatom as

filled circles, while all droplets that do not contain any *F. cylindrus* are displayed as open circles. This analysis reveals a

relatively sharp transition between filled and unfilled circles at $n_N$ values of about one ice nucleating active site per diatom

cell. All droplets frozen at $n_N \gtrsim 1$ (and lower temperatures) do not contain intact *F. cylindrus* diatoms. We suggest that their

freezing is induced by cell fragments or by INPs released by the *F. cylindrus* diatoms, e.g. soluble species from the EPS such

as proteins. A similar behaviour has been observed previously for birch pollen that release about $10^4$ ice nucleators per pollen

particle, which turned out to be ice-nucleating macromolecules (Pummer et al., 2012; Augustin et al., 2013; Pummer et al.,

2015; Dreischmeier et al., 2017).

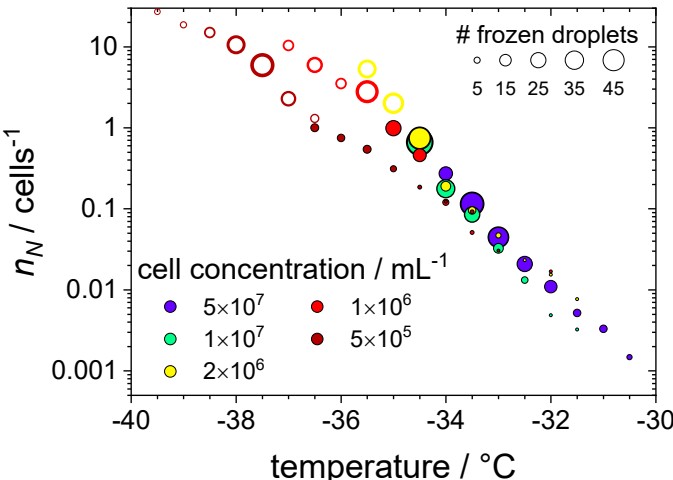

**Figure 6:** Cumulative number of ice nucleating sites $\boldsymbol{n_N}$ per number of *F. cylindrus* diatom cells as a function of temperature, obtained from the data shown in Fig. 5a with the help of Eq. (S2). The original data were binned into intervals of 0.5 °C. The size of the circle symbols
indicates the absolute number of droplets frozen in a particular bin, and the cell concentrations per mL are indicated by colour. The filled circles represent the droplets that contain whole *F. cylindrus* cells, while Poisson statistics suggest that the open circles should not contain any whole diatoms but probably some cell fragments, see text.

To verify the above interpretation, we performed experiments in which the samples from the measurements shown in Fig. 5a

and Fig. 6 were filtered with a pore size of 0.22 µm. This procedure should remove intact whole diatoms, whose size is about

4.5 to 74 µm for the apical axis and 2.4 to 4 µm for the transapical axis (Lundholm and Hasle, 2008; Cefarelli et al., 2010). In

Fig. 5b, the cumulative fraction of frozen droplets of these filtered samples is shown. The symbol colours represent the same

suspensions as shown in Fig. 5a, but this time filtered, and the artificial seawater reference data identical to that in panel (a).

All frozen fraction curves are shifted to lower temperatures when compared to the unfiltered samples, suggesting a significant





but not entire removal of INPs. Only the filtrate of the suspension with the smallest concentrations reveals a $T_{50}$ that is the
same as the seawater reference (-40.1 °C), suggesting that this sample does not contain any significant concentration of INPs
after filtration. All other filtrated suspensions show $T_{50}$ values that are higher by between 2.6 °C and 6.4 °C relative to the
seawater. For further information on the $T_{50}$ shifts, see Supplementary Table S4. Together these results imply that indeed either
fragments of *F. cylindrus* or molecules released by the diatoms can nucleate ice, but with a significantly reduced efficiency
than intact diatoms. Moreover, these results can also explain the observations in Fig. 6 of ice nucleation of droplets at $n_N \gtrsim 1$
that should not contain any diatoms. Below, we present further experiments to investigate the nature of the ice-nucleating
particles.

### 3.3 Ice nucleation of resuspended *F. cylindrus* cells

In the following experiments, we tried to separate diatoms from their fragments or any released INPs. For this purpose, the
sample suspension of *F. cylindrus* with a concentration of $10^7$ cells per mL, which was shown already in Fig. 5 above, was
analysed further, and the results are presented in Fig. 7. The green data points are those of the unfiltered sample and is identical
to that shown in Fig. 5a, and the magenta data points is identical to the filtered solution already presented in Fig. 5b (there as
green data points). This sample suspension should contain only INPs smaller than 0.22 µm. Next, most (but not all) of the
diatom cells and fragments contained in the filter cake of that filtration procedure were resuspended in artificial seawater.
Thus, the concentration of the resuspended cells is probably significantly smaller than $10^7$ cells per mL. The frozen fraction of
that sample is shown as the orange data points in Fig. 7 and shows the same onset ice nucleation temperature of about -32.5
°C as the original unfiltered suspension (green), however, the curve is much broader suggesting that it indeed contain much
less of the most active ice nucleators. In order to verify that all fragments smaller than 0.22 µm had been leached out during
the first filtration step, this resuspended filter cake sample was filtered again with a 0.22 µm filter. The results of this procedure
on the freezing behaviour is shown as the blue circles in Fig. 7. These frozen fraction data are practically identical to that of
the artificial seawater, strongly suggesting that indeed filtration of the pure whole cells had been successful and hardly any
fragments smaller than 0.22 µm are left in the filtrate. This analysis also implies that the ice nucleation of the unfiltered
suspension is due to whole cells as well as cell fragments, but not due to ice-nucleating molecules released from the diatoms.
The $T_{50}$ shift upon filtration of about 1.5 °C is similar in magnitude to the effect of reducing the concentration of the unfiltered
diatoms from $5 \times 10^7$ cells per mL to $1 \times 10^7$ cells per mL, i.e. by a factor of 5. This similarity may indicate that fragments make
up about 10-20% of the INPS in the unfiltered samples, which is in agreement with the fact that some ice nucleation is observed
for values of $n_N \gtrsim 1$, see Fig. 6.

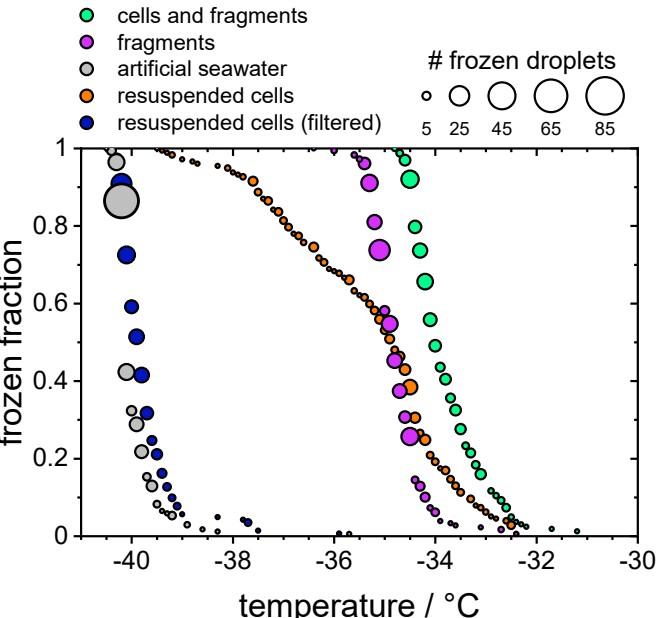

**Figure 7:** The frozen fraction of a sample with $10^7$ *F. cylindrus* diatoms per mL after different treatments. The symbol size indicates the total number of droplets frozen at that temperature. The green coloured data are the untreated sample and are the same as those in Fig. 5a.
The magenta data are the filtered sample that should just contain fragments of the diatoms. It is the same data as the green data in Fig. 5b. The grey data points show the freezing of the artificial seawater for reference (also replotted from Fig.5). The orange data show the freezing of the diatoms that were resuspended from the filter into artificial seawater. Its concentration is likely smaller than $10^7$ cells per mL, because not all cells could be resuspended. The blue data points represent the freezing of the droplets consisting of the resuspended cell suspension after renewed filtration: it should not contain any diatoms or fragments.

**3.4 Ice nucleation of spent medium and of purified *fc*IBP11**

We also investigated the spent *f*/2 medium (Guillard and Ryther, 1962), i.e., the medium in which the *F. cylindrus* diatoms were cultivated before they were separated by centrifugation to investigate their ice nucleating effects. Separation of the diatoms from the spent *f*/2 medium by centrifugation is not perfect and, hence, smaller fragments as well as soluble macromolecules such as proteins may remain in the spent medium. These may be potential ice nucleators, as it has been shown previously that even smaller ice-binding antifreeze proteins can act as ice nucleators at lower temperatures (Eickhoff et al.,
435 2019).

In Fig. 8 we compare the frozen fraction curve for the spent *f*/2 medium (light green circles) with that of a freshly prepared *f*/2 medium, which never had been in contact to any *F. cylindrus* diatoms (olive circles). Clearly, the spent medium, even after centrifuging off the diatoms, shows significant ice nucleation with a $T_{50}$ of about -35.7 °C, while the $T_{50}$ of the fresh medium is much lower at -40.0 °C. In additional experiments, the spent medium has been filtered in two further steps, first by using a
0.22 µm syringe filter (light blue circles) and then by using a 100 kDa centrifugation filter (pink circles). For comparison the fresh medium has been also filtered with a 100 kDa centrifugation filter (purple circles). Obviously, filtration of the spent





medium with a 0.22 µm filter shows hardly any effect on ice nucleation as its $T_{50}$ is shifted to -36.0 °C, which is the same as the unfiltered sample within the temperature uncertainty of our setup of ±0.3 °C.


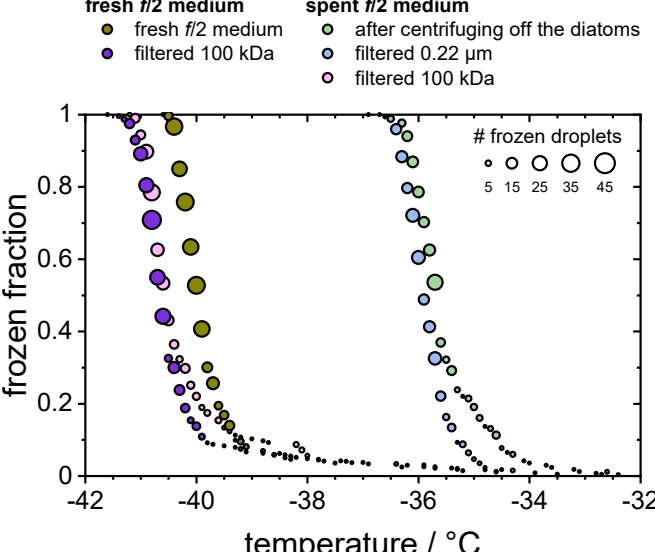

**Figure 8:** Frozen fraction of differently treated *f*/2 nutria media as a function of temperature. The olive and purple circles belong to a fresh *f*/2 medium that is untreated (olive) or had been filtered using a 100 kDa filter (purple). The green, blue and pink circles belong to the untreated, 0.22 µm filtered and 100 kDa filtered spent medium, in which the *F. cylindrus* diatoms had grown before they were centrifuged
and separated from the medium.

In contrast, filtration with a 100 kDa filter resulted in a strongly reduced ice nucleation with a $T_{50}$ value of -40.6 °C, which is the same as that of the filtrated fresh medium of -40.7 °C, suggesting that the 100 kDa filter removed all remaining ice nucleators present in the spent medium. This observation suggests that any macromolecules smaller than 100 kDa that were present in the spent medium are not ice nucleation active, because otherwise they had passed the filter and led to an increased

$T_{50}$ when compared to the fresh medium. The ice-binding proteins present in and/or released from *F. cylindrus* are similar in size to the well characterized *fc*IBP11, which is about 26 kDa (Bayer-Giraldi et al., 2011). Thus, ice-binding proteins released by the *F. cylindrus* into the spent medium should have passed the filter and could have induced ice nucleation, if they had significant ice nucleation activity. However, the results shown in Fig. 8 do not reveal any ice nucleation activity and, thus, can be interpreted as follows. Either, any proteins remaining in the filtrate do not promote ice nucleation or, alternatively, *F.*

*cylindrus* does not release any proteins into the spent medium. In order to shed further light on the ice-nucleating ability of ice-binding proteins from *F. cylindrus*, we studied purified *fc*IBP11 samples in additional experiments. We studied ice nucleation of two *fc*IBP11 solutions of different concentration as well as that of the pure Tris-HCl buffer for comparison. The results are presented in Fig. 9. The two *fc*IBP11 samples with concentrations of 0.1 mM (dark blue circles) and 0.01 mM (light blue circles) reveal $T_{50}$ values of -39.8 °C and -39.4 °C, which are equal to the $T_{50} = $ 39.7 °C of the buffer reference (black

circles) within experimental temperature uncertainty (±0.3 °C). Thus, no significant shift in the freezing temperature is





observed, and even when considering the increased ice nucleation temperature of the *fc*IBP11 at frozen fractions below about 25% it appears that *fc*IBP11 is not an efficient ice nucleator with relevance for atmospheric or biospheric processes, owing to its unnaturally high concentration in the droplet samples investigated here. These observations are in a good agreement with recent theoretical studies, which suggest that moderate IBPs show no nucleation of ice perpendicular to the basal and prismatic

ice planes (Cui et al., 2022). And indeed, these basal and prismatic planes are exactly those planes, at which the moderate fcIBP11 binds to ice (Kondo et al., 2018).

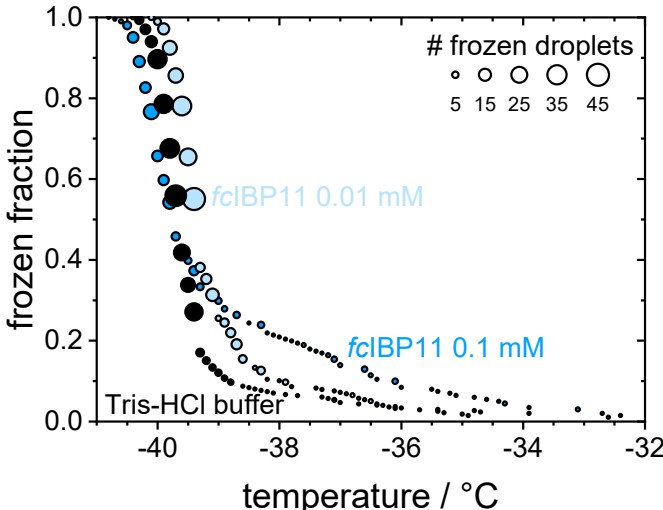

**Figure 9:** Cumulative frozen fractions as a function of temperature of droplets containing *fc*IBP11 solutions with concentrations of 0.1 mmol
per L (dark blue) and 0.01 mmol per L (light blue). The black circles show the freezing of the Tris-HCl buffer for reference. The circle area indicates the number of droplets frozen at a particular temperature.

Overall, the results show that *F. cylindrus* diatom cells as well as cell fragments suspended in seawater can induce heterogeneous ice nucleation, while ice-binding proteins produced by *F. cylindrus* such as *fc*IBP11 have negligible ice nucleation activity.

**4 Discussion and Implications**

Here, we put the results obtained above in the context of previous ice nucleation studies on diatoms. Triggered by the pioneering initial laboratory studies of marine diatom-induced ice nucleation (Alpert et al., 2011; Knopf et al., 2011) modelling studies have shown that in some regions of the atmosphere marine diatoms may indeed contribute to atmospheric INP (Burrows et al., 2013; Ickes et al., 2020). In order to use laboratory ice nucleation data in such models, the data have to be evaluated and
parameterized appropriately. For example, a direct comparison of $T_{50}$ or $f_{ice}$ originating from different laboratory studies on different types of INPs it not meaningful, as different sample volumes, INP concentrations, buffer concentrations, etc. may have been used. Therefore, it is preferable to compare the cumulative number of ice nucleating active sites per mass, surface




area or number of the INPs. Here, we make a comparison based on total INP mass, using the following definition of the cumulative number of ice nucleating active sites per mass $n_{m\_total}$ (Murray et al., 2012; Hiranuma et al., 2015; Hiranuma et al., 2019; Xi et al., 2021).

$$n_{m\_total} = \frac{-\ln(1-f_{ice})}{c_{m\_total} \cdot V} \qquad (8)$$

Here, $V$ is the volume of an individual droplet in the experiment and $c_{m\_total}$ is the total mass of biological material per droplet. For the *F. cylindrus* samples investigated here, we used the total carbon mass per *F. cylindrus* cell from the literature (Kang and Fryxell, 1992) and used elementary analysis to obtain the carbon content of our samples resulting in a value of 39.32 %. Using these values and our experimental data in Eq. (8), we have calculated the ice nucleating active sites $n_{m\_total}$ of the *F. cylindrus* diatoms, see the blue circles in Fig. 10. (We have fitted this data set and provide a corresponding parameterization, see Supplementary Fig. S6 and Eq. (S3).) Also shown in Fig. 10 are $n_{m\_total}$ data of other the sea ice diatoms *Melosira arctica* (blue squares) and *Nitzschia stellata* (blue triangles) and of the temperate diatom *Skeletonema marinoi* (open red circles) from previous studies (Ickes et al., 2020; Xi et al., 2021).

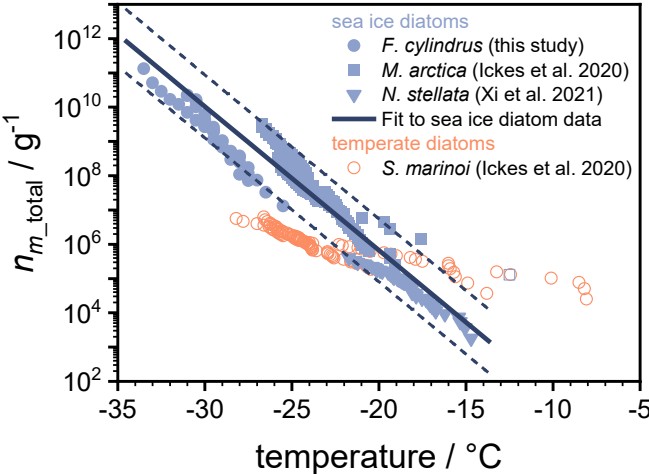

**Figure 10:** Experimental data of $\boldsymbol{n_{m\_total}}$, i.e. the number of ice active sites per total mass of *F. cylindrus* diatom cells (blue circles) and other sea ice diatoms (blue squares and triangles) from previous studies, as well as $\boldsymbol{n_{m\_total}}$ data for one temperate diatom species (open red circles) (Ickes et al., 2020; Xi et al., 2021). The solid line represents a fit of the $\boldsymbol{n_{m\_total}}$ values for the three sea ice diatom species (see Eq. (9)), while the dashed lines indicate the 2σ upper and lower prediction bands of this fit. All temperatures were corrected for the freezing point depressions of different buffers and solutes, so that they represent the ice nucleation induced by the diatoms in pure water.

The $n_{m\_total}$ values for *N. stellata* were provided by the authors (Xi et al., 2021). For *M. arctica* and the *S. marinoi*, we calculated $n_{m\_total}$ from the total number of cells given in the original work and provided by the authors (Ickes et al., 2020), and assume cell volumes of 653 μm³ and 125 μm³ and a cell density of 1 mg mL⁻¹ (Olenina et al., 2006; Xi et al., 2021). In order to allow a direct comparison of ice nucleation of the different diatoms, which were studied in different types of aqueous





solution, all the ice nucleation temperatures shown in Fig. 10 have been corrected for the colligative solute effect and represent diatom ice nucleation in pure water.

The comparison in Fig. 10 reveals that the curves of the three sea ice diatoms complement one another as $n_{m\_total}$ values of
different magnitude have been obtained over different temperature ranges. Interestingly, while there are some offsets between the different data sets, their slopes are quite similar. In contrast, the slope of the $n_{m\_total}$ data of the temperate diatom is significantly smaller. The observed similarities of the sea ice diatom data sets suggest a more generalized description of their behaviour in models. For this purpose, we fitted these data sets to provide a parametrization of $n_{m\_total}$ as a function of temperature. The three different data sets consist of different numbers of data points, which was taken into account in order to
give each data set the same statistical weight. We further note that one strongly deviating data point from the *M. arctica* data set (indicated as an open square in Fig. 10) was excluded from the fitting procedure. The resulting parameterization is given as:

$$\log_{10}(n_{m_{\text{total}}} \text{ g}^{-1}) = -0.420053 \,°C^{-1} \cdot T - 2.57818 \tag{9}$$


where $T$ is temperature to be entered in units of °C. For numerical code verification, Eq. (9) should result in a value for $n_{m_{\text{total}}}$ of $6.7 \times 10^5$ g$^{-1}$ at a temperature of -20.0 °C. This parametrization is valid over the temperature range between -13.7 °C to -34.5 °C (i.e., 259.45 to 238.65 K). The parameterization is shown as the thick solid line in Fig. 10, and the upper and lower 2σ prediction bands are given as dashed lines. In summary, Fig. 10 shows that the parameterization line and its prediction bands
are an appropriate representation of the ice nucleation activity of three types of sea ice diatoms suitable for use in atmospheric or biogeosciences model applications.

## 5 Conclusions

*F. cylindrus* diatoms can induce heterogeneous ice nucleation in artificial seawater by as much as 7.2°C higher than pure seawater for the highest diatom concentration investigated, ($5 \times 10^7$ cells per mL). We also observed an ice nucleating effect of
fragments smaller than 0.22 μm, in agreement with previous observations of the relevance of nanoscale biological fragments for ice nucleation in clouds (O'Sullivan et al., 2015; Wilson et al., 2015; Irish et al., 2017; Irish et al., 2019). We observed a common behaviour of the cumulative number of ice nucleating active sites per mass of diatom among three different types of sea ice diatoms. This similarity may originate from a similar biological function of the ice nucleation ability in seas-ice diatoms, and a corresponding parameterization developed thereof may simplify the representation of their properties in atmospheric
biogeoscientific models.

**Data availability**

The experimental data presented in this paper will be made freely available on a repository server of Bielefeld University upon final acceptance of the manuscript.

**Author contribution**

LE and TK designed the study. MBG provided the protein samples, LE performed the calibration and both the DSC and the microfluidic ice nucleation experiments, NR prepared the microfluidic devices. LE did the data analysis and the Poisson statistics calculations with input from TK. LE and TK prepared the figures, LE, TK and MBG wrote the manuscript with input from YR and NR. All authors contributed to the discussion of the data and text, and approved the final version of the manuscript.

**Competing interests**

The authors declare that they have no conflict of interest.

**Acknowledgements**

We thank Arika Allhusen and Klaus Valentin for providing the original *F. cylindrus* samples, and Luisa Ickes, Yu Xi and Allan Bertram for provision of original data sets on diatom ice nucleation and for helpful comments. We acknowledge support 555 for the publication costs by the Open Access Publication Fund of Bielefeld University and the Deutsche Forschungsgemeinschaft (DFG).

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
