# Peer review of "Ice nucleating properties of the sea ice diatom *Fragilariopsis cylindrus* and its exudates"

_Biogeosciences, 2022_

## Author Response (AR1)

Dear Editor,

below we provide a point-by-point response to all individual comments of the reviewers. We thank the reviewers for their constructive comments that helped improving the overall quality of our revised manuscript.

We indicate the reviewer's comments by black text, our answers are given in blue text, and new text that now appears in the revised manuscript is given in green text. In addition, we include at the end versions of the revised manuscript and the revised supplementary information with all changes indicated.

We hope that our revised manuscript now meets with your approval.

Sincerely yours,

Thomas Koop

**Answer to reviewer 1**

Eickhoff et al. present valuable results with a potential significant contribution to a better understanding of the microphysical processes in the ocean and the (polar) atmosphere.

However, in my opinion, some chapters could benefit from some shortening/rewriting, while the major findings should be rather put in a nutshell. Furthermore, I think, this manuscript is still lacking a look on the bigger picture and a critical discussion of its (atmospheric) significance and implications. See below for more specific comments.

We thank the reviewer for highlighting the importance of our contribution for the polar ocean and atmosphere and for the very detailed comments that have improved the revised version of the manuscript.

1. Introduction

L34-39 and L61-71.: Could you be more specific, which of the findings hold true for the Arctic, the Antarctic or both polar regions? This appears very important to me to not mix them up, since certain features of the Arctic and Antarctic are quite different.

The comments on the *Fragilariopsis cylindrus* hold for both the Artic and the Antarctic and we have updated that sentence accordingly.

The diatom *Fragilariopsis cylindrus* (see Fig. 1) is widespread in polar environments and is one of the predominant species within the Arctic and Antarctic microbial assemblages (Kang and Fryxell, 1992; Poulin et al., 2011; van Leeuwe et al., 2018).

Regarding atmospheric INP, the we have added the following new paragraph at the end of the introduction to discuss the differences in more detail:

There are some differences regarding the relevance of INPs in the Arctic and Antarctic polar regions. While in both polar latitudes the absolute concentrations of INPs are low, the influence of anthropogenic aerosols and INPs is much larger in the Arctic due to long-range transport during the Arctic winter (Šantl-Temkiv et al., 2019; Šantl-Temkiv et al., 2020; Ekman and Schmale, 2022). During the Arctic summer, aerosol lifetimes are shorter due to increased wet removal preventing long range transport and thus increasing the importance of locally emitted INPs. In the Antarctic, the influence of anthropogenic aerosols and INPs is generally much smaller (Stohl and Sodemann, 2010; Ekman and Schmale, 2022). During winter, blowing snow from the sea ice is the main aerosol source in the southern polar region, while DMS and other organic compounds from algae bloom are the main source during summer.

L34-35 Please add a reference.

The references were added as requested.

The diatom *Fragilariopsis cylindrus* (see Fig. 1) is widespread in polar environments and is one of the predominant species within the Arctic and Antarctic microbial assemblages (Kang and Fryxell, 1992; Poulin et al., 2011; van Leeuwe et al., 2018).

L40-42 Please recheck, if Wilson et al. (2015) or Aslam et al. (2018) are suitable references for supporting the statement of "the production of so-called ice-binding proteins (IBPs)"

The two references referred to only the second part of the sentence "and of other EPS also found in other diatom species". We have now added a reference for the first part of the sentence "the production of so-called ice-binding proteins (IBPs)" and separated the two parts with a comma:

One prominent example is the production of so-called ice-binding proteins (IBPs) (Bayer-Giraldi et al., 2011), and of other EPS that are also found in other diatom species (Wilson et al., 2015; Aslam et al., 2018).

L52-54 Which reference states that EPS do have good ice-binding properties? Is there any knowledge under which conditions (chemical composition) EPS do have these properties?

We have added the appropriate references and added the info that these EPS are mainly polysaccharides and proteins:

The very good ice-binding properties of *fc*IBP and EPS (mainly polysaccharides and proteins) under sea ice brine conditions have been reported in previous studies (Krembs et al., 2002; Bayer-Giraldi et al., 2011; Krembs et al., 2011).

L46-60 It might be important to clarify the relationship between the terms "ice-binding protein", "antifreeze protein" and "ice-nucleating protein" here. Do I get it right that less efficient ice-nucleating substances (in the sense that they are active at lower temperatures) can be at the same time antifreeze-substances at higher temperatures?

We have added some text to clarify the relationships of the different terms as requested:

L59-60 "Here we explore whether a similar ice-nucleating effect does occur also for IBPs from *F. cylindrus*" -> What is the outcome? This might be an important result that deserves to be mentioned in chapter "5. Summary and Conclusions"

The results of the ice nucleation experiments are now mentioned in the Summary:

For the ice-binding (antifreeze) protein *fc*IBP11, we did not observe any evidence for promoting ice nucleation at low temperatures.

2. Material and methods

L76 Please add the months, when ANT XVI/3 took place.

The information has been added:

The investigated *F. cylindrus* cells belong to the strain TM99 isolated in 1999 from the sea ice of the Weddel Sea, Antarctica, by Thomas Mock (*Polarstern* ANT XVI/3 expedition, which took place in the early spring from March to May 1999).

L76-81 When did the laboratory steps happen after the isolation in 1999? Back in 1999 or just recently, just before the ice nucleation experiments started? I just wonder how many cells (of those 108 cells) were still alive after a storage of approximately 20 years.

The cells were kept in culture since the expedition, the laboratory steps took place just recently, immediately before the experiments started. This information was added to the manuscript.

Since then, stock cultures were kept in *f*/2 medium (Guillard and Ryther, 1962) set up with Antarctic water and cultivated at 0°C and under continuous illumination of approximately 25 µE m$^{-2}$ s$^{-1}$. Before the experiment, cell numbers of the *F. cylindrus* cultures were monitored using a Coulter Counter, and cells were harvested during the exponential growth phase.

L84-94 This section could be shortened, since it contains much redundant information. It might be enough to refer to Tabel S1 in regard of the exact composition of the artificial sea salt.

Section has been shortened as suggested by removing the detailed information on the salts and referring to table S1.

The composition of the salts and their concentrations are given in Supplemental Information Table S1.

L 94, 100, 104, 110, 118, … If you only used one type of filter throughout this study, it might be enough to mention the filter type just once in the beginning.

We now mention the filter type only once with a more generalized statement:

The artificial seawater was filtered through a syringe filter (0.22 µm, Polyethersulfone, SimplePure) in order to exclude any effect of suspended dust particles on ice nucleation. This filter has been used for all filtrations in this study unless otherwise mentioned.

L103 I guess replacing "*F. cylindrus* cells" with "*F. cylindrus* samples" might make it more accurate.

Changed as suggested.

L106 Any reference that states that fcIBP11 is a soluble macromolecule detached from the cell? Or could it be connected to the cell surface of the diatoms as well?

According to Bayer-Giraldi et al., 2011, all analysed fcIBP11 isoforms are neither intracellular nor transmembrane, but secreted into the extracellular space. We have added the reference to the sentence.

L109-110 Is it possible to give an approximate estimate of the extend of cell loss?

We have added an approximate estimate as requested:

From the comparison of the frozen fraction curves obtained with the sample with those of unfiltered samples (see below) our best estimate of the concentration is about 2×10$^6$ cells per mL (estimated uncertainty range 1×10$^6$ – 1×10$^7$ cells per mL).

L136 "…belongs to the DUF3494 IBP family,…".. was already mentioned in the introduction (L 43). Why is it relevant to mention it here again?

We have removed the sentence in question as requested.

L145-184 Is it possible to shorten theses sections onto the relevant information since all these methods have been published before? Of course, the main principle and deviations from the original protocol should be mentioned. If you want to keep all details, maybe it is possible to shift part of these descriptions to the SI? Instead I would appreciate a short explanation, why you chose two different setups (2.3.1, 2.3.2) for the ice nucleation experiments.

We have shortened these paragraphs by shifting parts of the text to the SI. We have also added a short explanation, why we have used two different methods for our ice nucleation experiments:

The DSC experiment has been used as a simple and direct method to check whether *F. cylindrus* diatoms are potential ice nucleators or not. The method does not allow for the observation of single droplets, and we can only study cell fragments but not intact cells because the latter are disrupted during the emulsion preparation process. Therefore, we have used the WISDOM microfluidic device, which is described below, as the main experimental method in this study.

L185. You should check the appropriate use of "INP" throughout the manuscript. Maybe "ice nucleating molecules" or "ice nucleating entities" could be more correct here? I recommend to check the terminology proposed by (Vali et al., 2015).

We have used "INP", because we think "particle" is the most accurate term for the diatom cells and their fragments. This usage is also in line with many papers in the literature (including those of authors of the Vali et al. (2015) paper), in which diatoms were termed INPs, e.g.: (Creamean et al., 2021; McCluskey et al., 2018).

L186-188 Is this true? I think the methodology in the literature is just different to yours, where bigger droplets with higher volume per droplet were used. E.g. Budke and Koop (2015) used in the BINARY 1 µL droplets. Why did you choose for these small volumes of 380 pL (l 190) then?

We think there is a misunderstanding here. What we mean is that ~microliter droplet experiments usually employ a larger number of INPs per droplet (not INP concentration per volume). Moreover, these large droplet experiments are often subject to heterogeneous ice nucleation by impurities and the supporting surfaces at low temperatures, thus making it nearly impossible to study INPs with very low activity that nucleate only at temperatures just above homogeneous nucleation temperature. (The median freezing temperature of 'pure' water in the BINARY device is about -30°C, while the freezing temperature of the diatom experiments presented in Fig. 2 is between -32°C and -37°C.) For this purpose, small volume experiments such as the ~nanoliter microfluidic device used here are much more suitable, as they allow to reach homogeneous ice nucleation because the smaller droplets usually contain less impurities per droplet (The median freezing temperature of artificial seawater in the WISDOM device is about -40°C, see Fig.2). We have rewritten the text to make this clearer.

Ice nucleation studies using larger-volume droplet arrays usually employ relatively high concentrations of INPs per droplet, e.g. mineral dust particles or bacterial cells (Budke and Koop, 2015; Hiranuma et al., 2015; Wex et al., 2015; DeMott et al., 2018; Hiranuma et al., 2019; Kunert et al., 2019; Ickes et al., 2020), to ensure that freezing is induced at a temperature that is higher than that triggered by the

supporting surface or minute amounts of impurities contained in the water. In the present study, the total amount of INPs was small due to the limited availability of *F. cylindrus* cells, suggesting the use of small droplet methods which require less total INP material. We investigated droplets with a diameter of 90 µm, corresponding to a volume of about 380 pL. Another, probably more important advantage of using these small droplet volumes is that we can measure ice nucleation down to the homogenous freezing temperature of water (Riechers et al., 2013; Reicher et al., 2018; Tarn et al., 2021), enabling also the investigation of rather poor ice nucleators.

L189-192: I find this statement quite surprising, since you started your experiments with a solution of a highly concentrated culture (see 2.1). Then you performed several dilutions. And now you are stating that the number of INP was not enough for your experimental setup. Why did you perform dilutions then?

This question probably arises from the same misunderstanding discussed in the previous comment. We did these dilution experiments to obtain more accurate cumulative numbers of ice active sites per mass $n_{m\_total}$ and over a broader temperature range such that we can compare the data better to other literature data (see Figure 7).

L199-308 Is it possible to shift this section (or parts of it) into the SI?

As suggested, we have shifted the entire section (former lines L199-308) into the SI.

3. Results and Discussion
General:

Is it possible to bring this part more on a point? It feels like reading and rereading the same or similar aspects.

We have shortened this part by shifting the detailed description of DSC experiments including the former Fig.4 into the SI. We have also rewritten some sentences or parts and now term this section "Results" (see your comment on section names further below).

L 326 How can freezing at -44°C be still relevant, when water droplets usually freeze at -38°C homogeneously? Is it because of the small volume of the droplets in your setup?

The droplets in the DSC experiments (and also in the microfluidic experiments) are all large enough not to be affected by the Gibbs-Thomson effect. The homogeneous freezing temperature of water droplets in the DSC experiments is indeed -38°C, as defined by the onset of the exothermic signal. The corresponding onset temperatures for the two samples, sea-water and sea-water containing *F. cylindrus*, are shifted to lower temperatures (-39°C to-40°C) due to the colligative freezing point depression effect of the salts contained in the seawater. We added this information to this text (now in the SI):

Because of the colligative freezing point depression of the seawater, the freezing temperatures of the reference and the sample are shifted to lower temperatures, compared to pure water.

Figure 5: Just as another example, where sentences can be shortened. Instead of: "b): Freezing temperatures of the same samples shown in panel (a), but after filtration with a pore size of 0.22 μm." you could write: "b): Freezing temperatures of the filtered (0.22 μm) samples."

Changed as suggested.

Line 369 "all diatoms" (?) or all cells?

We agree and changed the term to 'all *F. cylindrus* cells' to make this point more clear. We also made corresponding changes at several other places throughout the text where appropriate.

L378 Within the whole manuscript, proteins are proposed as likely ice nucleating molecules. What about polysaccharides? Any tests performed into this direction?

Yes, it is known from the literature that polysaccharides can also act as ice-nucleating molecules (e.g. Dreischmeier et al. (2017)), so we added the term polysaccharides to the sentence in question:

We suggest that their freezing is induced by cell fragments or by INPs released by the *F. cylindrus* diatoms, e.g. soluble species from the EPS such as proteins or polysaccharides.

However, as the experiments on soluble components described further below in the paper did not reveal any ice nucleating effect, we did not analyse our samples for polysaccharide content.

L379 "macromolecules" or maybe "polysaccharides", to be more specific. Why is the comparison of diatoms with birch pollen relevant here?

We used the comparison to birch pollen here just to mention that by filtering the *F. cylindrus* samples, it is entirely possible that ice nucleation activity can remain, when either smaller fragments or soluble ice-nucleating molecules passes through the filter, thus resulting in n_N values larger than 1. This is exactly what happens when pollen suspended in water are filtered and the remaining washing water contains ice nucleating molecules. Also in the case of pollen, the n_N values reach values far above 1 (about $10^4$ per pollen grain to be precise).

4. Discussion and Implications

General:

What is the difference between the chapters "3. Results and Discussion" and "4. Discussion and Implications"? It appears to me that the text (or at least parts) of chapter 4 in the current version of the manuscript might still represent a subchapter of "3. Results and Discussion".

As suggested we have renamed chapters 3-5 and also shifted some of their contents. Chapter 3 is now simply termed "Results". Chapter 4 remains "Discussion and Implications" because we compare our results to those of other diatoms and combine these datasets into a single parameterization. Moreover, we added two new paragraphs to discuss implications of our results for the atmosphere and compare them to (the very sparse) Southern Ocean diatom and INP measurements (see your next comments).

This study was mainly motivated with the atmospheric relevance of INPs (e.g lines 64-72, lines 534-536). However, a critical discussion of these (new) findings for atmospheric implications are still missing and could fit here. Some of the following aspects should be discussed in this section:

Which atmospheric residence time would you expect for diatoms, their fragments and exudates? Whole marine diatoms are rather big and might precipitate within few seconds or minutes. Can it be expected that complete cells/fragments will make it into the atmospheric layers relevant for mixed-phase clouds or even cirrus clouds? (as implied in lines 66-70)

We have added some of this information to the text.

Which are the atmospheric concentrations of diatoms/*Fragilariopsis cyclindus* or fragments in the ambient air?

As far as we now there are no such measurements.

In Figure 10, you nicely compare the ice nucleating activity of *Fragilariopsis cyclindus* with several diatoms from other studies. However, a rating of the importance of diatoms as INP in the polar regions/Southern Ocean in comparison to other types of INP (such as marine bacteria, mineral dust, …) in regard of abundance and/or ice nucleating efficiency is missing.

The Antarctic is known for a low number of efficient atmospheric INPs in comparison to the Arctic (e.g. (McCluskey et al., 2018; Wex et al., 2019; Hartmann et al., 2021; Zeppenfeld et al., 2021)). Considering this fact, how would you evaluate the importance of your findings for a better understanding of the Antarctic environment/atmosphere?

We have added two new paragraphs relating our data on *F. cyclindus* and sea ice diatoms to marine INP data and also try to compare the potential importance of our experimental data to Southern Ocean INP abundances. As there are only very sparse data on diatoms and INP in the Southern Ocean and the Antarctic marine environment these can only be regarded as order of magnitude estimates. Clearly, more field experiments in these regions are highly desirable. The new text reads:

In the following, we put the ice nucleation data of *F. cylindrus* and the other sea ice diatoms into context by comparing to field studies. Wilson et al. (2015) provided experimental evidence for a marine biogenic source of ice nucleating particles and suggested that exudates and fragments of diatoms as a source of the ice nucleating material located in the sea surface microlayer. Their low-temperature freezing data reveals a cumulative number of ice nucleating active sites per total organic carbon mass $n_{m\_TOC}$ of ~$1.3 \times 10^{10}$ g$^{-1}$ at -27 °C (calculated from the equation given in the caption of their Fig. 2), which is the low-temperature end of their data, and the most relevant to the present study. To compare this value to the $n_{m\_total}$ values given in Fig. 7, we estimated that the organic carbon content of their samples varies between 39.32% (representing the organic carbon content of *F. cylindrus* cells, see above) or 100% (representing a purely organic carbon composition), resulting in a range of $n_{m\_total}$ of ~$5.0 \times 10^{9}$-$1.3 \times 10^{10}$ g$^{-1}$ for their Arctic sea surface microlayer samples. These are compared to $n_{m\_total}$ values of $8.2 \times 10^{7}$ g$^{-1}$ (2σ prediction bands: $2.8 \times 10^{7}$-$2.4 \times 10^{8}$ g$^{-1}$) for *F. cylindrus* and of $5.8 \times 10^{8}$ g$^{-1}$ (2σ prediction bands: $7.0 \times 10^{7}$-$4.8 \times 10^{9}$ g$^{-1}$) for sea ice diatoms, respectively, at -27°C, indicating that *F. cylindrus* and other sea ice diatoms may contribute to the marine INP in the Southern Oceans and Antarctic seawater, assuming the Wilson et al. parameterization applies also to these areas.

In another comparison, we use measurements of insoluble aerosol particles made at Amsterdam Island in the Southern Indian Ocean (Gaudichet et al., 1989). These measurements show that marine biogenic particles make up between 8 and 28% of the number of detected particles and that these were predominantly assigned to *Radiolaria* and diatom fragments (identified as amorphous silicates), with about 27 % or $2.7 \times 10^4$ m$^{-3}$ particles observed in the southern winter (July) and fewer in fall (May, 8 %, $2.4 \times 10^4$ m$^{-3}$) and spring (September, 7 %, $1.8 \times 10^3$ m$^{-3}$). If we assume that all *Radiolaria* and diatom fragments can be attributed to *F. cylindrus* diatoms, we can calculate the mass concentration of *F. cylindrus* diatom cells per cubic meter of air from the mass per individual cell ($m_{total}$ = $4.5 \times 10^{-11}$ g, see above), yielding values of $1.2 \times 10^{-6}$ g m$^{-3}$ air (July), $1.1 \times 10^{-6}$ g m$^{-3}$ air (May), and $8.1 \times 10^{-8}$ g m$^{-3}$ air (September). Using the parametrization of the cumulative number of ice nucleating active sites per mass *F. cylindrus* in Eq. (S10), we calculate a $n_{m\_total}$ value of $8.2 \times 10^7$ g$^{-1}$ (2σ prediction bands: $2.8 \times 10^7$-$2.4 \times 10^8$ g$^{-1}$) at -27 °C, see above, from which we can derive the ~88 INP m$^{-3}$ air (2σ: 3-250) at -27 °C in fall (May) . This value can be compared to *in situ* total INP measurements in the Southern Ocean south of Australia in fall (March-April) yielding values between 34 and 207 INP m$^{-3}$ air at -27 °C (McCluskey et al., 2018). Although the above calculations are order of magnitude estimates , the comparison shows that it is not unreasonable that sea ice diatoms such as *F. cylindrus* and their fragments may constitute a significant fraction of the INP in the Southern Ocean and Antarctic waters.

Specific:

L494 You performed own elementary analysis for obtaining the carbon content in your samples? Could you please add the method to chapter "2. Materials and Methods"? Few lines might be sufficient.

As requested, we have added a few lines on the elemental analysis in the new section 2.4:

The total carbon content of the *F. cylindrus* samples has been determined using elemental analysis. For this purpose, an amount of 0.7 mg *F. cylindrus* diatoms was combusted at a high temperature ($T > 1000$ °C) in a Tin-crucible and the composition was analysed using a commercially available elemental analyser (EuroVector, Euro EA).

Figure 10: Is it anyhow possible to still extend this figure with the experimental freezing results on the diatom *Thalassiosira pseudonana* by (Wilson et al., 2015) or (Knopf et al., 2011)?

Knopf et al. (2011) were the first to show that marine diatoms can act as INPs. However, as they presented only freezing temperature data, but no n_m or n_N values, we could not include these data in our comparison in Figure 7 (former Fig.10)

The same is true for the *Thalassiosira pseudonana* data in Wilson et al., 2015. Nevertheless, we now make a numerical comparison in the text to the sea surface microlayer freezing data of Wilson et al., 2015 at -27°C, which is the low temperature end for the cumulative number of INPs of their study (their Fig.2b) and the high temperature end of our *F. cylindrus* data. Note, that the direct comparison is difficult, as the provide their data as cumulative number of INPs per gram of **total organic carbon** (TOC). As the source of the ice nucleator and the species distribution in the sea surface microlayer is not known to us, we had to estimate the mass fraction of the organic carbon. The related text now reads:

Wilson et al. (2015) provided experimental evidence for a marine biogenic source of ice nucleating particles and suggested that exudates and fragments of diatoms as a source of the ice nucleating material located in the sea surface microlayer. Their low-temperature freezing data reveals a cumulative number of ice nucleating active sites per total organic carbon mass $n_{m\_TOC}$ of ~1.3x10$^{10}$ g$^{-1}$ at -27 °C (calculated from the equation given in the caption of their Fig. 2), which is the low-temperature end of their data, and the most relevant to the present study. To compare this value to the $n_{m\_total}$ values given in Fig. 7, we estimated that the organic carbon content of their samples varies between 39.32% (representing the organic carbon content of *F. cylindrus* cells, see above) or 100% (representing a purely organic carbon composition), resulting in a range of $n_{m\_total}$ of ~5.0x10$^9$-1.3x10$^{10}$ g$^{-1}$ for their Arctic sea surface microlayer samples. These are compared to $n_{m\_total}$ values of 8.2x10$^7$ g$^{-1}$ (2σ prediction bands: 2.8x10$^7$-2.4x10$^8$ g$^{-1}$) for *F. cylindrus* and of 5.8x10$^8$ g$^{-1}$ (2σ prediction bands: 7.0x10$^7$-4.8x10$^9$ g$^{-1}$) for sea ice diatoms, respectively, at -27°C, indicating that *F. cylindrus* and other sea ice diatoms may contribute to the marine INP in the Southern Oceans and Antarctic seawater, assuming the Wilson et al. parameterization applies also to these areas.

Figure 10: Now you show values which are normalized on mass (total mass?) At which part did you include the carbon content (L 494-495) then?

Originally, we only know the number of diatom cells per mL of water, which is why we calculated the cumulative number $n\_N$ of ice nucleating sites per number of *F. cylindrus* diatoms. In the literature, many studies (including field studies) provide the cumulative number $n\_N$ of ice nucleating sites per mass. Hence, to compare our freezing data to other studies in the literature the mass concentration of the *F. cylindrus* diatoms was required and, thus, the mass per individual *F. cylindrus* diatom cell $m_{total}$. However, in the literature we only found a value for the mass of carbon per *F. cylindrus* diatom. Hence, we needed the mass fraction of carbon in the *F. cylindrus* diatoms (which we could determine experimentally, see new section 2.4). To make this procedure more clear, we added the calculation of $m_{total}$ in the manuscript:

For the *F. cylindrus* samples investigated here, we used the total carbon mass per *F. cylindrus* cell from the literature (Kang and Fryxell, 1992) and performed elemental analysis to obtain the carbon content of our samples, resulting in a value of 39.32 % to calculate the average total mass per individual *F. cylindrus* diatom cell of $m_{total}$ = 4.5x10$^{-11}$ g. Using these values and our experimental data in Eq. (1), we have calculated the ice nucleating active sites $n_{m\_total}$ of the *F. cylindrus* diatoms, see the blue circles in Fig. 7.

L505-507: "All temperatures were corrected for the freezing point depressions of different buffers…" How did you do it? Did you follow the approach by Koop and Zobrist (2009)?

For the data from Xi et al. (2021), we only received freezing temperature data that were already corrected for the freezing point depression by these authors. Hence, we have used their data for our comparison plot.
For the data from Ickes et al., we got their raw data with both the uncorrected freezing temperatures and those corrected for the freezing point depression. Hence, we have used their corrected ones for our comparison plot.
For our own data, we have obtained the freezing point depression by directly measuring the shift in ice nucleation temperature between pure distilled water and the artificial seawater that we used. We have corrected all measured nucleation temperatures of *F. cylindrus* diatoms in seawater by this temperature shift. We have added the following text:

To allow a direct comparison of ice nucleation of the different diatoms, which were studied in different types of aqueous solutions, all the ice nucleation temperatures shown in Fig. 7 have been corrected

(either by the original authors or by us) for the colligative solute effect and represent diatom ice nucleation in pure water. We have corrected the freezing temperatures of the *F. cylindrus* samples by the measured difference between the $T_{50}$ of pure double-distilled water and pure artificial seawater without any diatoms.

L508-510 Is it necessary to mention this in the main text? It could be sufficient to add this information as a footnote in Figure 10.

Changed as suggested.

L528 Is there any reason, why you convert °C to K at this late part of the manuscript? You have not done it before, so why here?

We removed the additional information on the Kelvin temperature range to avoid confusion.

5.Conclusions
General:

The current version of the text rather represents a summary of the manuscript. However, conclusions are still sparse in this section. I'd recommend adding some real conclusions and a renaming of this chapter "Summary" or "Summary and Conclusions".

We have renamed this Chapter to "Summary and Conclusions" as requested and also made some text changes to that part.

Minor comments:

L67 "can be transported"

We inserted missing word "be".

L538: Remove "s" from "seas-ice"

The extra letter has been removed.

L538 Check for a consistent writing of "sea-ice diatoms" versus "sea ice diatoms" throughout the manuscript

Checked and unified to „sea ice diatoms".

**Answer to reviewer 2**

This is a sound set of experiments showing an increase of 7.2 °C in the ice nucleation temperatures for seawater containing F. cylindrus diatoms when compared to pure seawater. The laboratory study seem carried out well and the literature review is state of the art. There are two important aspect to be considered before this paper can be accepted.

We thank the reviewer for this positive evaluation of our study and for the detailed comments that has improved the revised version of the manuscript.

The paper does not read well and it seems a bit too long. I suggest to merge the results discussion implication or to short by half all the text in the last two section. Once have a feeling there are many sentences saying the same and not really giving a clear simple message. Clean up and make a simple clear concise message.

We have significantly shortened the manuscript by deleting several parts of the Materials and Methods (subsections 2.3.1, 2.3.2, and nearly entirely 2.3.3 including Figs. 2 and 3) and also the entire subsection 3.1.1 of the Experimental Results (including Fig. 4) and moving them into the Supplementary Information. Furthermore, we have edited the sections 3. Results, 4. Discussions and Implications, and 5. Summary and Conclusions. Following suggestions by reviewer 1, we renamed these sections and also added some text on the atmospheric relevance of our experimental results on the ice nucleation of *F. cylindrus* diatom cells and fragments, as well as sea ice diatoms in general to section 4.

It seems to be that in the abstract and conclusion, and also in the introduction (well written) one of the main result is the results "that *F. cylindrus* diatom cells as well as cell fragments suspended in seawater can induce heterogeneous ice nucleation, while icebinding proteins produced by *F. cylindrus* such as *fc*IBP11 have negligible ice nucleation activity.". This is important and also compared with the literature, but what is the reason? Any literature support any speculation and possible reasons? This is in stark contrast with other literature supporting the idea of proteins being important in INP, but little is discuss in the text of this paper. I suggest to expand this extensively given it seems a major result. It is also important to give possible biogeochemical reasons of cell fragments being more important than proteins.

We have now stated this result also in the Summary and Conclusion section. The reason why some ice-binding 'antifreeze' proteins act as at least moderate ice nucleators (such as the *Tenebrio molitor tm*AFP) and others show only minute or now ice nucleation activity (such as the *fc*IBP11 studied here) is not entirely clear, but a recent modelling study suggest that this may have to do with the ice planes that they usually bind to (see statement at the end of section 3). However, typical ice-nucleating proteins are usually much larger than these ice-binding 'antifreeze' proteins and their large ice-active site may therefore be much better suited in supporting a newly forming ice embryo. We have enhanced the section explaining the differences and similarities between ice-binding proteins, antifreeze proteins and ice-nucleating proteins that may also help understanding these phenomena:

[revised manuscript text omitted]

The larger broad signal in both emulsion samples corresponds to the homogeneous ice nucleation temperature of artificial seawater. This signal is also observed in the *F. cylindrus* sample because many of the emulsion droplets in that sample do not contain diatoms. The exothermic shoulder of the signal, which is not present in the reference, is most likely due to the freezing

of droplets containing a diatom cell or fragment, and the shift of the onset to higher temperature is a first indication for the heterogeneous ice nucleation activity of the diatoms.

Because of the fact that the diatoms are of similar size as the emulsion droplets and the potential of mechanical disruption of diatom cells during the fast stirring of the disperser during emulsion preparation, these emulsion experiments appear to us as not suitable for a quantitative analysis of the ice nucleation activity of *F. cylindrus*. Thus, we employed non-invasive methods in the experiments described below.

**Parametrization of *F. cylindrus* ice nucleation efficiency**

In Eq. (2) in the main paper, we provide a parametrization representing the ice nucleation of the different sea ice diatoms in shown in Fig. 7 of the main paper in terms of the number of ice active sites per total mass of diatom cells, $n_{m\_total}$. We also derived a parametrization for the individual ice nucleation efficiency of the *F. cylindrus* diatoms (see Fig. S9), which is given in the following Eq. (S10):

$$\log_{10}(n_{m\_total} \text{ g}^{-1}) = -0.521789°\text{C}^{-1} \cdot T - 6.1761. \qquad\qquad \text{(S10)}$$

where $T$ is temperature to be entered in units of °C. For numerical code verification, Eq. (S10) should result in a value for $n_{m\_total}$ of $8.2 \times 10^7$ g$^{-1}$ at a temperature of -27.0 °C. This parametrization is valid over the temperature range between -24.5 °C to -34.5 °C.

**Table S1:** Salts used for the preparation of artificial seawater for the *F. cylindrus* ice nucleation experiments. The amounts of substances provided for each ion yield a mass of 500 g artificial seawater at a salinity of 34.5.

| Salt | Supplier | $m$ [g] | $Na^+$ [mmol] | $K^+$ [mmol] | $Mg^{2+}$ [mmol] | $Ca^{2+}$ [mmol] | $Cl^-$ [mmol] | $SO_4^{2-}$ [mmol] | $H_2O$ [mmol] |
|---|---|---|---|---|---|---|---|---|---|
| NaCl | VWR Chemicals | 11.8446 | 202.68 | | | | 202.68 | | |
| KCl | VWR Chemicals | 0.3758 | | 5.04 | | | 5.04 | | |
| $MgCl_2 \cdot 6H_2O$ | ITW Reagents | 5.3280 | | | 26.21 | | 52.42 | | 157.25 |
| $Na_2SO_4 \cdot 10H_2O$ | Acros Organics | 4.4902 | 27.87 | | | | | 13.94 | 139.36 |
| $CaCl_2 \cdot 2H_2O$ | ITW Reagents | 0.7460 | | | | 5.07 | 10.15 | | 10.15 |
| $H_2O$ | double distilled water | 477.23 | | | | | | | 26490.26 |
| artificial seawater | | **500.01** | **230.55** | **5.04** | **26.21** | **5.07** | **270.28** | **13.94** | **26797.02** |

**Table S2:** Temperature parameters used in the microfluidic freezing experiments. The first number in each triplet is the final temperature of the respective step in °C, the second number indicates the rate of cooling or heating in °C per min, and the third number indicates the holding time at the final temperature in min. Reference samples were always investigated with the same parameters as those given for each sample.

| Step | *F. cylindrus* | *F. cylindrus* (filtered) | *F. cylindrus* (pure cells) | *F. cylindrus* (Medium) | *fc*IBP11 | *P. syringae* |
|---|---|---|---|---|---|---|
| 1 | -20/-5/2 | -20/-5/2 | -20/-5/2 | -20/-5/2 | -20/-5/2 | -5/-5/2 |
| 2 | -45/-1/0 | -45/-1/0 | -45/-1/0 | -45/-1/0 | -45/-1/0 | -40/-1/0 |
| 3 | -10/5/2 | -10/5/2 | -10/5/2 | -10/5/2 | -10/5/2 | -10/5/2 |
| 4 | 5/1/0 | 5/1/0 | 5/1/0 | 5/1/0 | 5/1/0 | 5/1/0 |

**Table S3:** As prepared concentrations $c$ of the *P. syringae* samples, calculated fractions of droplets with at least one bacterium $P_\lambda(k \geq 1)_{\text{calculated}}$, as well as measured fractions $P_\lambda(k \geq 1)_{\text{measured}}$ and experimentally determined concentrations $c_{\text{measured}}$ based on the approach outlined above using Eq. (S8).

| $c$ / mL$^{-1}$ | $P_\lambda(k \geq 1)$ calculated | $P_\lambda(k \geq 1)$ measured | $c$ measured / mL$^{-1}$ |
|---|---|---|---|
| 1.4x10$^7$ | $1.00^{+0.00}_{-0.01}$ | 0.99 | 1.2x10$^7$ |
| 2.8x10$^6$ | $0.66^{+0.06}_{-0.06}$ | 0.61 | 2.5x10$^6$ |
| 1.4x10$^6$ | $0.41^{+0.05}_{-0.05}$ | 0.39 | 1.3x10$^6$ |

**Table S4:** Shifts in ice nucleation temperature relative to the $\Delta T_{50}$ of artificial seawater for the untreated *F. cylindrus* samples, as well as for the samples filtered with a 0.22 µm syringe filter.

| $c$ | unfiltered $\Delta T_{50}$ | filtered $\Delta T_{50}$ |
|---|---|---|
| $5 \times 10^7 \text{ mL}^{-1}$ | 7.2 °C | 6.4 °C |
| $1 \times 10^7 \text{ mL}^{-1}$ | 6.0 °C | 5.2 °C |
| $2 \times 10^6 \text{ mL}^{-1}$ | 5.4 °C | 3.1 °C |
| $1 \times 10^6 \text{ mL}^{-1}$ | 4.8 °C | 2.6 °C |
| $5 \times 10^5 \text{ mL}^{-1}$ | 2.8 °C | 0.0 °C |

[Figure]

**Figure S1:** Extraction of the pure *F. cylindrus* cells by filtration of the stock solution (green). After filtration, the filtrate (purple) should only contain smaller cell fragments and soluble molecules such as *fc*IBP, while whole cells and larger fragments remain on the filter (orange filter). By shaking the filter in artificial seawater (grey), the cells were resuspended (orange solution). As a finally test, filtration of this suspension (blue) should not show any ice nucleation results different from those of pure artificial seawater.

**spent *f*/2 medium**

[Figure]

filter 0.22 μm

centrifuge
filter 100 kDa

spent *f*/2 medium
(after centrifuging off the diatoms)

spent *f*/2 medium
filtered 0.22 μm

*f*/2 medium
filtered 100 kDa

**fresh *f*/2 medium**

[Figure]

centrifuge

filter 100 kDa

fresh *f*/2 medium

*f*/2 medium
filtered 100 kDa

**Figure S2:** Sample preparation for the ice nucleation experiments with the *f*/2 medium. The spent medium should only contain a few diatoms, because the diatoms were separated from the medium by centrifugation before (green vial). By filtration with a syringe-filter, we removed the remaining cells and retained smaller *F. cylindrus* fragments and the *fc*IBP in the filtrate (blue solution). The solution was filtered by centrifugation filtration and the resulting filtrate should only contain soluble macromolecules smaller than 100 kDa, e.g. *fc*IBP (pink vial). The fresh *f*/2 medium (olive solution) does not contain any cells, fragments or *fc*IBP and was also filtered by centrifugation filtration as a reference (purple vial).

[revised manuscript text omitted]

---

## Author Response (AR2)

Dear Dr. Anja Engel,

Thank you very much for accepting our manuscript subject to technical corrections. Below we provide our response to the editorial comments as well as to those of Reviewer 1.

The comments are given in black text, our answers in blue, and new text in the technically revised version in green. In addition, we include the revised manuscript and supplement with all the very minor and technical changes indicated.

Sincerely yours,

Thomas Koop
* * *
**Changes following editorial suggestions:**

1.) When preparing the files for final publication please assure that all references are included.

We have checked that all references were included.

2.) Moreover, please check the annotation for units for consistence.

We have harmonized the notation of the units as requested by reviewer 1.

3.) Merve Parla from editorial office noted: Please ensure that the colour schemes used in your maps and charts allow readers with colour vision deficiencies to correctly interpret your findings.

Following this advice, we have modified Figures 2-5 by adding small black dots in the center of the green symbols in order to make them better distinguishable from reddish colors for readers with color vision deficiencies.

4.) Data availability:

We have added the original data of the figures as a data table to the supplement and modified the data availability statement accordingly:

The experimental data presented in the figures of this paper are provided in tabular form in the supplement.

**Answers to Reviewer 1 and related changes:**

a) Section 3.1. "The results are described in detail in the Supplemental Information and in Fig. S8." (Line 210). This is the first result of your manuscript. I guess you should start with strong findings that are presented in the main manuscript. Why did you decide to shift this important section to the SI? The reviewers asked you to shorten the sentences, not to shift the results to the SI.

In the first round of the review process, the reviewers wrote: "*some chapters could benefit from some shortening/rewriting, while the major findings should be rather put in a nutshell*" and "*3. Results and Discussion General: Is it possible to bring this part more on a point?*" and "*The paper does not read well and it seems a bit too long. I suggest to merge the results discussion implication or to short by half all the text in the last two section.*"

Following these suggestions, we have removed the paragraphs describing the results of the DSC experiments (including the former Fig.4) and shifting them into the SI, because we find these results are not essential. They indeed show that there is a difference in the freezing behaviour between seawater and seawater with diatoms, but they cannot be used to quantify the ice nucleation activity. Therefore, we have put the focus on the quantitative results obtained with the microfluidics setup.

b) L84-91 In my opinion, this (new) last chapter of the introduction appears a bit misplaced to me and several references (e.g. L87f, L89ff) are missing.

We added this paragraph following the request by Reviewer 1 during the first round of reviews: "L34-39 and L61-71.: Could you be more specific, which of the findings hold true for the Arctic, the Antarctic or both polar regions? This appears very important to me to not mix them up, since certain features of the Arctic and Antarctic are quite different."

We have added references at the end of the two sentences (L87f and L89ff) as requested.

If the authors want to keep it, I wouldn't object. However, I recommend to replace it with a transitional paragraph about the objectives of your study and how it will contribute to a better understanding of atmospheric INP.

We prefer to keep the paragraph, but we have added the following transitional paragraph on the objectives of our study as requested by the reviewer:

In the following, we present experimental data on the ice nucleation activity of F. cylindrus diatom cells and their exudates. We then analyse and convert these data into a quantifiable format so that they can be compared to other measurements of this type. Finally, we provide a comparison to ice nucleation data of other polar diatoms together with a parameterization that generalizes their ice nucleation activity for use in atmospheric models.

General: sometimes you use the "per" notation (e.g. L 149, 150, 151), sometimes you write "-1"(e.g. L157). Maybe you like to harmonize throughout the manuscript?

We thank the reviewer for this suggestion. We have removed the term "per" wherever possible, and throughout the manuscript and supplement we converted all units to the exponential notation, i.e., $g^{-1}$, $mL^{-1}$, $m^{-3}$ etc.

[revised manuscript text omitted]
, while the highest accuracy can be reached for a value of 50 % (see blue curve in Fig. S5a and Fig. S6). For higher concentrations, when every droplet contains at least one INP, the above Poisson evaluation is not needed and the classic method can be used, and so this upper limit sets an endpoint for the Poisson-based evaluation. The classic method indeed assumes that every observed droplet contains at least one INP and it has been described in detail previously (Murray et al., 2012; Budke and Koop, 2015).

To demonstrate the concentration range suitable for the Poisson method, i.e. the Poisson relevant range, the latter is indicated in Fig. S5a as the grey shaded area. The solid blue curve shows the values of $P_\lambda(k \geq 1)$ calculated using Eq. (S7) as a function of the average INP concentration $c$ of the studied sample and a droplet diameter of 90 µm. The two dashed lines show the changes for a deviation of $\pm 5$ µm in droplet diameter.

To verify the procedure, we investigated aqueous suspensions of the well-studied ice-nucleating bacterium *Pseudomonas syringae* in the form of the commercial product Snomax (Morris et al., 2011; Budke and Koop, 2015; Wex et al., 2015). The ice nucleation temperatures of each about 165$\pm$15 droplets, from three single measurements with 45 to 70 droplets each, containing either pure double-distilled water or three different concentrations of *P. syringae* were investigated, see Table S3. These concentrations are also marked in Fig. S5a as vertical lines. A similar plot for the *F. cylindrus* diatoms can be found in Fig. S6. The resulting experimental frozen fraction curves of *P. syringae* are shown in Fig. S5b. Double-distilled water (black open symbols) shows a steep increase in frozen fraction below about -34.0 °C, in agreement with homogeneous ice nucleation rates of droplets of such diameter (Koop and Murray, 2016; Reicher et al., 2018; Eickhoff et al., 2019). Following this observation, all droplets of the *P. syringae* samples that froze at around or below this temperature are assumed to have nucleated homogenously, i.e. they are considered to contain no INPs in the analysis below.

For all *P. syringae* samples, the first freezing events occur at much higher temperatures of about -8 to -9 °C, and the frozen fraction curve in each case initially increases strongly before reaching a plateau, and subsequently the remaining liquid droplets freeze only at very low temperatures. In each sample, the plateau occurs at a different value of the frozen fraction, e.g. $f'_{\text{ice}}$ is higher the larger the *P. syringae* concentrations (pink > blue > orange). We determined the corresponding $f'_{\text{ice}}$ values, as defined above, from the experimentally obtained frozen fraction curve as the value of the frozen fraction at -34.0 °C, i.e. at the threshold between heterogeneous and homogenous ice nucleation as defined above. The resulting $f'_{\text{ice}}$ values for the three concentrations were 0.99, 0.61, and 0.39, respectively, indicated as the dashed horizontal lines in Fig. S5b. These $f'_{\text{ice}}$ values correspond to $P_\lambda(k \geq 1)_{\text{measured}}$ and can be used to infer the average INP concentration from Eq. (S7). Because in the current experiments the INP concentrations are known (i.e., $1.4 \times 10^7$, $2.8 \times 10^6$. and $1.4 \times 10^6$ mL$^{-1}$), these experimentally derived $f'_{\text{ice}}$ values can be compared to the expected $f_{\text{ice}}$ values, corresponding to $P_\lambda(k \geq 1)_{\text{calcualted}}$ values calculated from Eq. (S7), yielding values of 1.00$\pm$0.01, 0.66$\pm$0.06 and 0.41$\pm$0.05, respectively. These theoretical values are in good agreement (within experimental uncertainty) with the measured values and thus confirm our approach and the inferred INP concentrations of $1.2 \times 10^7$, $2.5 \times 10^6$ and $1.3 \times 10^6$ mL$^{-1}$ (see Table S3) deviate by about 14%, 11% and 7% from the prepared concentrations, which is very good given that INP concentrations can vary by orders of magnitude. For further validation that the Poisson distribution is necessary for a proper evaluation in the above-mentioned concentration range, the cumulative number of active ice-nucleating sites $n_N$ per number of *P. syringae* bacteria was evaluated and discussed in the following section and the related Fig. S7.

**Determination of INP concentration**

Above, we have defined $f'_{ice}$ as the plateau region separating heterogeneous and homogenous freezing. Since $f'_{ice}$ varies with the number of droplets containing at least one INP, an experimentally determined $f'_{ice}$ value can be used to calculate the concentration of INPs for unknown samples using a variation of Eq. (S7). Typically, a sample is investigated by means of a dilution series so that a different INP concentration is scanned in each experiment. If the INP concentration is too large, all droplets freeze heterogeneously, and if it is too low, no INP-induced heterogeneous nucleation occurs (apart from that induced by any impurity present) and, thus, all droplets freeze homogeneously. In both these cases, it is not possible to obtain the desired INP concentration. But if measurements are done in the Poisson relevant concentration range, one can observe both heterogeneous as well as homogenous freezing of droplets, resulting in a plateau in the frozen fraction curve, as discussed above. With the frozen fraction value of this plateau, $f'_{ice}$, and the assumptions that, first, every INP induces heterogeneous freezing and that, secondly, all heterogeneously frozen droplets freeze before the first freezing of a homogenous frozen droplet, the following equation can be obtained by rearranging Eq. (S7):

$$c = -\frac{6\ln\left(1-P_\lambda(k\geq1)\right)}{\pi\cdot d^3} = -\frac{6\ln\left(1-f'_{ice}\right)}{\pi\cdot d^3}. \tag{S8}$$

[revised manuscript text omitted]

**Parametrization of *F. cylindrus* ice nucleation efficiency**

In Eq. (2) in the main paper, we provide a parametrization representing the ice nucleation of the different sea ice diatoms in shown in Fig. 7 of the main paper in terms of the number of ice active sites per total mass of diatom cells, $n_{m\_total}$. We also derived a parametrization for the individual ice nucleation efficiency of the *F. cylindrus* diatoms (see Fig. S9), which is given in the following Eq. (S10):

$$\log_{10}(n_{m\_total}\ \text{g}^{-1}) = -0.521789°\text{C}^{-1} \cdot T - 6.1761.\hspace{2cm}(\text{S10})$$

where $T$ is temperature to be entered in units of °C. For numerical code verification, Eq. (S10) should result in a value for $n_{m\_total}$ of $8.2 \times 10^7$ g$^{-1}$ at a temperature of -27.0 °C. This parametrization is valid over the temperature range between -24.5 °C to -34.5 °C.

**Table S1:** Salts used for the preparation of artificial seawater for the *F. cylindrus* ice nucleation experiments. The amounts of substances provided for each ion yield a mass of 500 g artificial seawater at a salinity of 34.5.

| Salt | Supplier | $m$ [g] | $Na^+$ [mmol] | $K^+$ [mmol] | $Mg^{2+}$ [mmol] | $Ca^{2+}$ [mmol] | $Cl^-$ [mmol] | $SO_4^{2-}$ [mmol] | $H_2O$ [mmol] |
|---|---|---|---|---|---|---|---|---|---|
| NaCl | VWR Chemicals | 11.8446 | 202.68 | | | | 202.68 | | |
| KCl | VWR Chemicals | 0.3758 | | 5.04 | | | 5.04 | | |
| $MgCl_2 \cdot 6H_2O$ | ITW Reagents | 5.3280 | | | 26.21 | | 52.42 | | 157.25 |
| $Na_2SO_4 \cdot 10H_2O$ | Acros Organics | 4.4902 | 27.87 | | | | | 13.94 | 139.36 |
| $CaCl_2 \cdot 2H_2O$ | ITW Reagents | 0.7460 | | | | 5.07 | 10.15 | | 10.15 |
| $H_2O$ | double distilled water | 477.23 | | | | | | | 26490.26 |
| **artificial seawater** | | **500.01** | **230.55** | **5.04** | **26.21** | **5.07** | **270.28** | **13.94** | **26797.02** |

**Table S2:** Temperature parameters used in the microfluidic freezing experiments. The first number in each triplet is the final temperature of the respective step in °C, the second number indicates the rate of cooling or heating in °C  min, and the third number indicates the holding time at the final temperature in min. Reference samples were always investigated with the same parameters as those given for each sample.

| Step | F. cylindrus | F. cylindrus (filtered) | F. cylindrus (pure cells) | F. cylindrus (Medium) | fcIBP11 | P. syringae |
|------|--------------|--------------------------|----------------------------|------------------------|---------|-------------|
| 1 | -20/-5/2 | -20/-5/2 | -20/-5/2 | -20/-5/2 | -20/-5/2 | -5/-5/2 |
| 2 | -45/-1/0 | -45/-1/0 | -45/-1/0 | -45/-1/0 | -45/-1/0 | -40/-1/0 |
| 3 | -10/5/2 | -10/5/2 | -10/5/2 | -10/5/2 | -10/5/2 | -10/5/2 |
| 4 | 5/1/0 | 5/1/0 | 5/1/0 | 5/1/0 | 5/1/0 | 5/1/0 |

**Table S3:** As prepared concentrations $c$ of the *P. syringae* samples, calculated fractions of droplets with at least one bacterium $P_\lambda(k \geq 1)_{\text{calculated}}$, as well as measured fractions $P_\lambda(k \geq 1)_{\text{measured}}$ and experimentally determined concentrations $c_{\textbf{measured}}$ based on the approach outlined above using Eq. (S8).

| $c$ / mL$^{-1}$ | $P_\lambda(k \geq 1)_{\text{calculated}}$ | $P_\lambda(k \geq 1)_{\text{measured}}$ | $c_{\text{measured}}$ / mL$^{-1}$ |
|:---:|:---:|:---:|:---:|
| $1.4 \times 10^7$ | $1.00^{+0.00}_{-0.01}$ | 0.99 | $1.2 \times 10^7$ |
| $2.8 \times 10^6$ | $0.66^{+0.06}_{-0.06}$ | 0.61 | $2.5 \times 10^6$ |
| $1.4 \times 10^6$ | $0.41^{+0.05}_{-0.05}$ | 0.39 | $1.3 \times 10^6$ |

**Table S4:** Shifts in ice nucleation temperature relative to the $\Delta T_{50}$ of artificial seawater for the untreated *F. cylindrus* samples, as well as for the samples filtered with a 0.22 µm syringe filter.

| $c$ | unfiltered $\Delta T_{50}$ | filtered $\Delta T_{50}$ |
|---|---|---|
| $5\times10^7$ mL$^{-1}$ | 7.2 °C | 6.4 °C |
| $1\times10^7$ mL$^{-1}$ | 6.0 °C | 5.2 °C |
| $2\times10^6$ mL$^{-1}$ | 5.4 °C | 3.1 °C |
| $1\times10^6$ mL$^{-1}$ | 4.8 °C | 2.6 °C |
| $5\times10^5$ mL$^{-1}$ | 2.8 °C | 0.0 °C |

[Figure]

**Figure S1:** Extraction of the pure *F. cylindrus* cells by filtration of the stock solution (green). After filtration, the filtrate (purple) should only contain smaller cell fragments and soluble molecules such as *fc*IBP, while whole cells and larger fragments remain on the filter (orange filter). By shaking the filter in artificial seawater (grey), the cells were resuspended (orange solution). As a finally test, filtration of this suspension (blue) should not show any ice nucleation results different from those of pure artificial seawater.

**spent _f_/2 medium**

[Figure]

filter 0.22 μm

centrifuge
filter 100 kDa

spent _f_/2 medium
(after centrifuging off the diatoms)

spent _f_/2 medium
filtered 0.22 μm

_f_/2 medium
filtered 100 kDa

**fresh _f_/2 medium**

[Figure]

centrifuge
filter 100 kDa

fresh _f_/2 medium

_f_/2 medium
filtered 100 kDa

[revised manuscript text omitted]